# ExpVid: A Benchmark for Experiment Video Understanding & Reasoning

Yicheng Xu[1,2*]   Yue Wu[1*]   Jiashuo Yu[1]   Ziang Yan[1]   Tianxiang Jiang[1]   Yinan He[1]
Qingsong Zhao[1]   Kai Chen[1]   Yu Qiao[1]   Limin Wang[1,3]   Manabu Okumura[2]   Yi Wang[1†]
[1]Shanghai AI Laboratory   [2]Institute of Science Tokyo   [3]Nanjing University
yxu040@e.ntu.edu.sg, wangyi@pjlab.org.cn

## Abstract

Multimodal Large Language Models (MLLMs) hold promise for accelerating scientific discovery by interpreting complex experimental procedures. However, their true capabilities are poorly understood, as existing benchmarks neglect the fine-grained and long-horizon nature of authentic laboratory work, especially in wet-lab settings. To bridge this gap, we introduce ExpVid, the first benchmark designed to systematically evaluate MLLMs on scientific experiment videos. Curated from peer-reviewed video publications, ExpVid features a new three-level task hierarchy that mirrors the scientific process: (1) Fine-grained Perception of tools, materials, and actions; (2) Procedural Understanding of step order and completeness; and (3) Scientific Reasoning that connects the full experiment to its published conclusions. Our vision-centric annotation pipeline, combining automated generation with multi-disciplinary expert validation, ensures that tasks require visual grounding. We evaluate 20 leading MLLMs on ExpVid and find that while they excel at coarse-grained recognition, they struggle with disambiguating fine details, tracking state changes over time, and linking experimental procedures to scientific outcomes. Our results reveal a notable performance gap between proprietary and open-source models, particularly in high-order reasoning. ExpVid not only provides a diagnostic tool but also charts a roadmap for developing MLLMs capable of becoming trustworthy partners in scientific experimentation. Data are publicly available at `https://github.com/OpenGVLab/ExpVid`.

## 1 Introduction

Scientific progress is driven by careful experimentation. In wet-lab settings such as biology, chemistry, and medicine, researchers need to execute fine-grained actions with exacting precision, adhere to stepwise protocols, and reason from procedures to results (Gabrieli et al., 2025; Yagi et al., 2025). Yet understanding and reproducing these procedures are time-consuming for practitioners and opaque to newcomers. Recent advances in Multimodal Large Language Models (MLLMs) (OpenAI, 2025; DeepMind, 2025b; Bai et al., 2025c) make it tempting to delegate parts of this workflow to artificial intelligence: perceiving experimental manipulations, checking procedural fidelity, and even connecting observed operations to scientific conclusions. Regarding this, a question remains: how well do current MLLMs understand real experimental footage?

Despite steady progress on video-based benchmarks (Li et al., 2024a; Hu et al., 2025; Hasson et al., 2025), most existing datasets emphasize general actions or activities or medical computer vision scenarios rather than authentic laboratory experimentation. These settings lack the distinctive challenge of wet-lab work: visually subtle operations (e.g., pipetting microliter volumes), small and often occluded tools, fine-grained materials and states, and long-horizon dependencies that link early preparation steps to downstream results. To our knowledge, there is no systematic evaluation targeting the spectrum of capabilities needed for assisting research from operational perception through procedural understanding to higher-order scientific analysis in genuine experiment videos.

---

[*]Equal contribution.
[†]Corresponding author.

We introduce ExpVid, a benchmark for scientific experiment video understanding and reasoning. It spans 13 disciplines and centers on wet-lab experiments; a small number of dry-lab or field engineering videos are included for breadth and completeness, while purely computational and most physics experiments are excluded. Each video is paired with a peer-reviewed publication to ensure scientific rigor and to support annotations linking video experiments to innovations and conclusions. In term of sources, ExpVid is curated from online peer-reviewed research collection (JoVE), whose exo-view recordings capture real-world laboratory manipulations with detailed narration.

To assess models across both temporal and analysis difficulty granularity, ExpVid organizes data into three tiers: single-step perceptions within seconds, multi-step understanding over minutes, and full-experiment as scientific reasoning across extended workflows. In this regard, we define a task hierarchy that mirrors how scientists work. At the operational level, models must recognize tools, materials, quantities, and fine-grained actions in short clips. At the procedural level, models predict over stage-level segments by ordering steps, verifying completeness, and predicting next moves. At the reasoning level, models integrate visual evidence across the full video and relate it to the accompanying paper to answer questions about motivation, significance, and conclusions.

Specifically, we adopt a vision-centric annotation method to generate viable question–answer pairs at multiple temporal scales, and then introduce human expertise to secure the correctness. Questions are constructed so that visual cues, instead of background knowledge alone, are necessary, along with carefully designed distractors that are semantically and visually plausible. Multidisciplinary experts then validate, refine, and balance the items to ensure domain fidelity and diversity across disciplines and procedures. This combination of automated construction and expert verification yields a relatively scalable yet rigorous benchmark tailored to the realities of experimental science.

We use ExpVid to evaluate 20 popular MLLMs (with both open-source and proprietary). The findings (in Sec. 4) reveal clear strengths in coarse object recognition and short-horizon reasoning, but persistent challenges in (i) disambiguating visually similar tools and materials under occlusion, (ii) tracking quantities and states across steps, and (iii) connecting procedural evidence to scientifically valid conclusions. These also emphasize that reliable visual grounding and structured reasoning are most urgently needed in real laboratory settings (mostly wet-lab tasks). We believe these chart a roadmap for MLLM research toward trustworthy assistants or agents that can perceive, verify, and reason about real experiments rather than stylized demonstrations.

In summary, our contributions are given as:

- We present ExpVid, to our best knowledge, the first benchmark that systematically evaluates MLLMs on scientific experimental footage across three hierarchical levels: fine-grained perception, procedural understanding, and scientific reasoning.

- We design a scalable vision-centric annotation pipeline that constructs multi-level tasks from videos, associated ASR transcripts and peer-reviewed papers, followed by rigorous multidisciplinary expert validation and refinement.

- We benchmark 20 leading MLLMs on ExpVid and provide an analysis of our results. We show ExpVid can work as a foundation for measuring and advancing MLLMs in real laboratory settings.

## 2 RELATED WORK

**Multimodal Large Language Models (MLLMs).** MLLMs extend LLMs to multimodal domains by combining visual perception with linguistic reasoning. Both closed-source models (e.g., GPT-5 (OpenAI, 2025), Gemini 2.5 Pro (DeepMind, 2025b)) and open-source models (Chen et al., 2024b; Zhu et al.; Bai et al., 2025c; Hong et al., 2025) demonstrate strong reasoning capabilities on multimodal inputs. Some further address ultra-long video understanding, enabling reasoning over hours of content (Bai et al., 2025c; Wang et al., 2025b; Li et al., 2024b). To advance scientific discovery, Intern-S1 (Bai et al., 2025a) is tailored for scientific domains. Nevertheless, MLLMs' ability to understand and reason over laboratory experiment videos remains underexplored.

**Video understanding benchmarks.** Existing video benchmarks evaluate video models on general video understanding tasks, including for example, action recognition (Caba Heilbron et al., 2015; Sigurdsson et al., 2016; Mangalam et al., 2023), dense captioning (Das et al., 2013; Rohrbach et al., 2015; Chai et al., 2024), and temporal grounding (Gao et al., 2017; Lei et al., 2021; Liu et al., 2024).

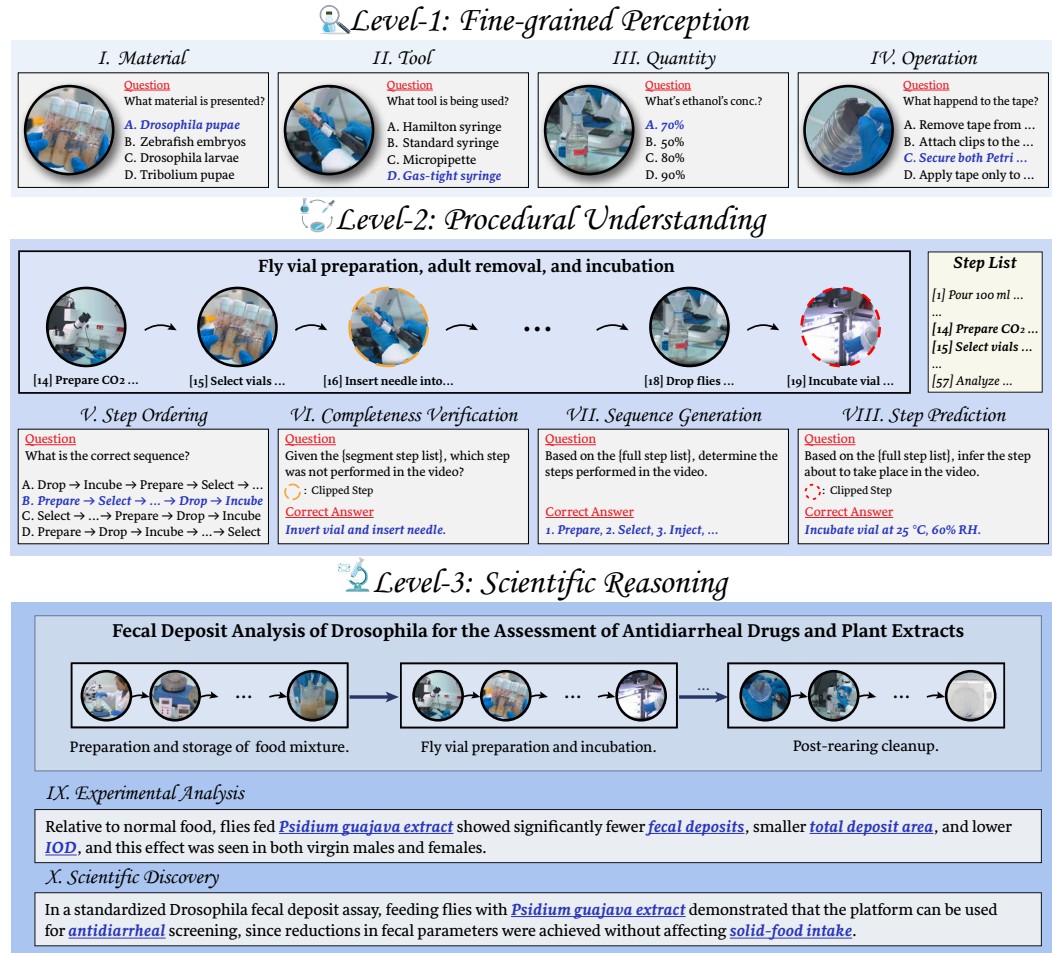

Figure 1: Illustration of three-level task hierarchy in ExpVid.

Video-MME (Fu et al., 2025) and MVBench (Li et al., 2024a) provide comprehensive evaluations on short video clips with multi-choice questions, while several works such as MLVU (Zhou et al., 2024), LVBench (Wang et al., 2024c), VRBench (Yu et al., 2025), evaluate MLLMs on long video comprehension or introduce narrative-driven dataset for multi-step reasoning in extended video contexts. These benchmarks advance perception and temporal reasoning, but remain agnostic to domain-specific scientific knowledge and experimental contexts.

**Knowledge-driven and scientific benchmarks.** Another stream of work emphasizes knowledge-intensive evaluation, requiring models to integrate discipline knowledge beyond perception. Chem-Bench (Alampara et al., 2025), MathVision (Wang et al., 2024b), and MathVista (Lu et al., 2023) are for specific domains. Broader efforts (Yue et al., 2024; Zhao et al., 2025; Wang et al., 2024d; Chen et al., 2024a) target expert-level, multi-disciplinary tasks, with Video-MMMU (Hu et al., 2025) extending this to domain knowledge from videos. Recently, SCI-VID (Hasson et al., 2025) and SFE (Zhou et al., 2025) further introduce scientific benchmarks, but focus on outcome recognition (e.g., medical images), rather than understanding whole experiments. Yet real-world scientific discovery critically depends on lab experiments, where step-wise operations and tools drive results.

## 3 EXPVID: A SCIENTIFIC EXPERIMENT VIDEO BENCHMARK

We develop a benchmark to assess the performance of MLLMs on experimental footage. Specifically, we mostly focus on wet experiments related to biology, chemistry and medicine. Only a few dry ones (e.g., field engineering) are included while most computational examples, or examples from

Table 1: Comparison between ExpVid and some MLLM benchmarks. A and M indicate automatic and manual annotation, respectively.

| Benchmark | #QA Pairs | #Videos | Avg. Sec. | #Task Types | Annotation | Domain |
|---|---|---|---|---|---|---|
| *General Video Benchmarks* | | | | | | |
| MVBench (Li et al., 2024a) | 4,000 | 3,641 | 16.0 | 20 | A+M | General |
| Video-MME (Fu et al., 2025) | 2,700 | 900 | 1,017.9 | 1 | M | General |
| MLVU (Zhou et al., 2024) | 3,102 | 1,730 | 930.0 | 9 | M | Narrative |
| VRBench (Yu et al., 2025) | 9,468 | 960 | 5,796.0 | 1 | M | Narrative |
| *Knowledge-driven Benchmarks* | | | | | | |
| MMVU (Zhao et al., 2025) | 3,000 | 1,529 | 51.4 | 2 | M | Multi-disc. |
| Video-MMMU (Hu et al., 2025) | 900 | 300 | 506.2 | 3 | M | Multi-disc. |
| MathVision (Wang et al., 2024b) | 3,040 | – | – | 1 | M | Math |
| MathVista (Lu et al., 2023) | 6,141 | – | – | 31 | M | Math |
| MMMU (Yue et al., 2024) | 11,500 | – | – | 2 | M | Multi-disc. |
| ScienceQA (Saikh et al., 2022) | 21,208 | – | – | 1 | M | Science |
| SciBench (Wang et al., 2023) | 789 | – | – | 1 | M | Science |
| MMStar (Chen et al., 2024a) | 1,500 | – | – | 6 | M | Multi-disc. |
| SFE (Zhou et al., 2025) | 830 | – | – | 66 | M | Science |
| **ExpVid** | 7,800 | 390 | 489.0 | 10 | A+M | Science |

physics are excluded. Since wet experiments commonly own higher operational costs and complexity than dry ones, they demand more in intelligent assistance and analysis. In the following, we first describe ExpVid's data curation (Sec. 3.1), then present its task hierarchy (Sec. 3.2) and finally detail the annotation (Sec. 3.3). An overview of the benchmark construction pipeline is illustrated in Fig. 2.

## 3.1 EXPERIMENT DATA CURATION

**Collection.** We collect scientific experiment videos, automatic speech recognition (ASR) transcripts, and corresponding papers from the Research section of JoVE (Journal of Visualized Experiments), a multi-disciplinary, peer-reviewed video journal. JoVE publishes step-by-step experimental protocols in video format, allowing viewers to observe the fine-grained manipulations and precise procedures. Its exo-view recordings of lab experiments yield high-quality visual content, while associated ASR transcripts offer detailed procedural descriptions, which are well-suited for annotation. The paired peer-reviewed papers further allow us to design challenging reasoning tasks that bridge experimental procedures to research conclusions and scientific findings.

**Filtering.** For quality control, we apply a multi-dimensional scoring process to ASR transcripts via DeepSeek-R1 (Guo et al., 2025a). Each transcript is rated on five criteria (0-5 scale): 1) **Continuity**: whether covers the video without temporal gaps or missing segments. 2) **Alignment**: whether its timestamps align with the actual video duration; 3) **Clarity**: its logical coherence, domain-appropriate terminology, and overall readability; 4) **Integrity**: whether it records an entire experimental workflow, including distinct procedural stages; 5) **Focus**: whether centers on procedures rather than background, lectures, or unrelated context.

An overall score is obtained by averaging across five dimensions, and only those scored at least 4 overall with no dimension below 3.5 are retained, yielding a high-quality subset. Additionally, videos are constrained to the interquartile range of durations (25th–75th percentiles, 378–728s) to remove outliers. Within each scientific discipline, experiments are ranked by overall scores and manually reviewed to exclude videos that predominantly feature computer-screen displays or lack actual laboratory footage. Further, multi-disciplinary experts select 30 top-ranked experiments from each of the 13 disciplines, yielding 390 videos with ASR transcripts averaging 1,026 words. This ensures ExpVid remains balanced and diverse. Detailed statistics, along with the list of all 13 disciplines, are reported in Appendix B.1.

**Preprocessing.** For a systematical evaluation across temporal scales, we process all videos into a three-level hierarchy to probe distinct capabilities.

- **Level-1: Action-level Clips.** We obtain ∼10k clip-text pairs (with each lasting ∼8s on average). Specifically, we segment ASR transcripts by punctuation and align each sentence with its times-

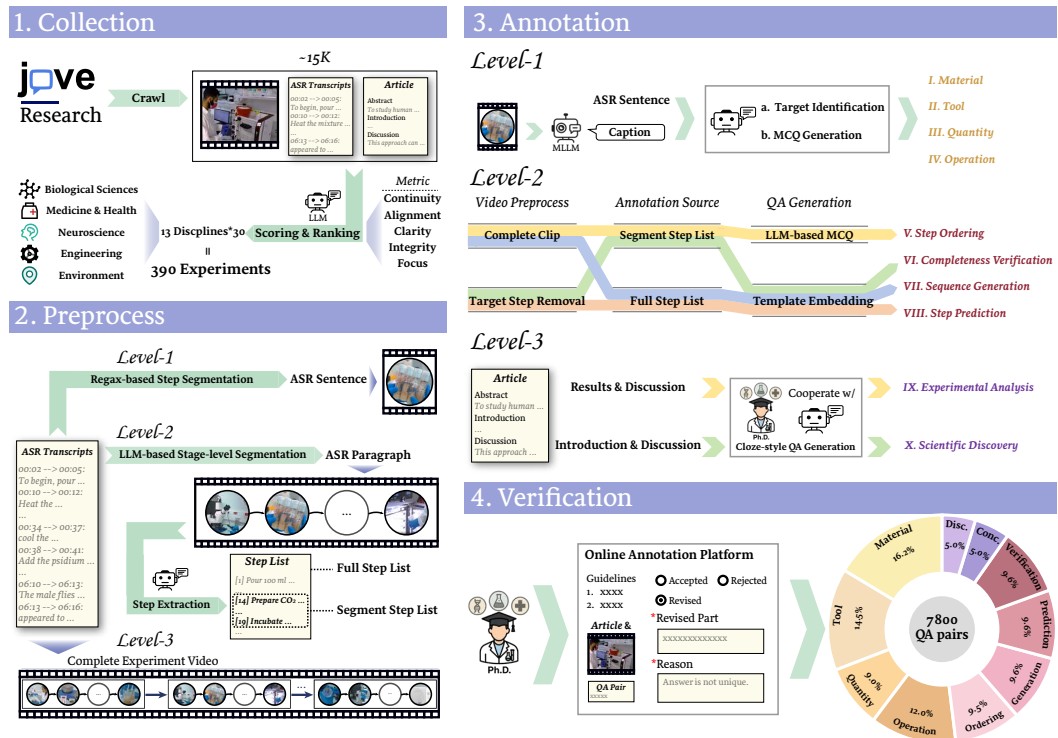

Figure 2: An overview of ExpVid construction pipeline.

tamp to cut the video. This yields clip–ASR sentence pairs that provide step-wise experimental narrations, well-suited for perception-oriented tasks such as action or material recognition.

- **Level-2: Stage-level Segments.** We get ∼3.5k segment-text pairs with an average duration of ∼48s. We divide each experiment into semantically coherent stages (e.g., preparation, main procedures, post handling). We use DeepSeek-R1 to generate stage-level boundaries for each ASR transcript, guided by prompts that enforces both logical and causal continuity across operations. Each ASR paragraph is constrained to 20–60s to preserve temporal coherence while avoiding excessive context length. From each paragraph, DeepSeek-R1 further extracts step-level operation descriptions to form a segment step list. Concatenating all segment step lists reconstructs a full step list, which serves a suitable basis for procedural understanding tasks.

- **Level-3: Full Procedure Videos.** We directly preserve the full experiment videos (average ∼8 minutes). In certain cases, we remove concluding slides, figures, and data-analysis segments to avoid potential shortcuts (e.g., models exploiting textual conclusions) and ensure evaluation relies on procedural content. This level targets long visual context and structural reasoning, requiring models to integrate information across extended experimental workflows.

## 3.2 TASK HIERARCHY IN EXPVID

Based on the processed videos of varied lengths, we define ExpVid's three-level task hierarchy, benchmarking MLLMs on scientific experiment videos, ranging from short-term perception to long-term reasoning. This design allows us to progressively evaluate models' abilities: whether they can recognize fine-grained visual details, predict over coherent experimental procedures, and ultimately reason scientific conclusions over lab experiments. Fig. 1 illustrates this hierarchy.

**Level-1: Fine-grained Perception.** It evaluates whether MLLMs can visually ground essential elements in short clips of individual steps through four Multi-Choice Question (MCQ) tasks:

- **Material Recognition:** Distinguish the target experimental material and distinguish it from other plausible substances commonly encountered in laboratory settings.

- **Tool Recognition:** Identify the tools that appear in the scene and reject visually or functionally similar distractors.

- **Quantity Recognition:** Choose the correct numerical attribute (e.g., *dosage*, *temperature*) by visually interpreting scales, amounts, or counts.

- **Operation Recognition:** Recognize the specific action being performed in the video and differentiate it from confusable but incorrect operations in the similar setup (e.g., *Insert → Attach*).

**Level-2: Procedural Understanding.**   This type of task evaluate models on their reasoning about logical and temporal order across multiple steps within stage-level clips, including:

- **Step Ordering:** Select the correct step execution order when the original sequence is perturbed into plausible but incorrect arrangements.

- **Sequence Generation:** Given the candidates, find out the ordered steps that appear in the clip.

- **Completeness Verification:** Given the candidates, detect the missing step in the clip.

- **Step Prediction:** Given the first $n - 1$ steps of an experiment stage, predict the next step $n$.

**Level-3: Scientific Reasoning.**   It has two tasks that require models to integrate visual experiment processes with domain knowledge to draw conclusions, in the form of fill-in-the-blank questions:

- **Experimental Analysis:** Infer crucial conclusions from experimental data, e.g., compare current results with existing studies, highlight new findings, and explain the corresponding mechanisms.

- **Scientific Discovery:** Reason over the entire experiment video, move beyond current outcomes, and abstract broader insights, such as linking results or innovations to larger scientific phenomena, interpreting the significance in filling blanks of which domain or potential application values, and proposing improved solutions for the current limitations and new directions for this area.

### 3.3   VISION-CENTRIC ANNOTATION WITH KNOWLEDGE GUIDANCE

Our annotation pipeline adopts a semi-automatic strategy that combines LLM assistance with human expert verification. To ensure benchmark *vision-centric*, we deliberately avoid encoding contextual cues from the narration that could directly reveal answers during QA construction. Moreover, distractors are crafted to be semantically or visually plausible, forcing models to rely on visuals rather than purely leveraging LLM priors and textual cues. To minimize the LLM bias, LLM is limited to extracting experimental entities (e.g., subjects, actions, tools) from ASR transcripts and transforming them into QA candidates. Human experts then review, refine, and validate these annotations for correctness. Building upon the hierarchy given in Sec. 3.1 and 3.2, we construct them as follows.

**Fine-grained Perception.**   For the four perception tasks *Material*, *Tool*, *Quantity*, and *Operation*, candidate entities or actions are first extracted from ASR sentences by DeepSeek-R1 as targets and aligned with video clips, with a Qwen2.5-VL captioner providing visual triggers to verify their visibility. Normalization preserves critical states of materials and essential identifiers of tools, while excluding under-specified or generic terms. Then, these resulting targets are converted into four-option multiple-choice questions (MCQs), where distractors are generated by DeepSeek-R1 following task-specific prompt rules: for *Material* and *Tool*, distractors reflect visual/functional similarity or common confusions; for *Quantity*, they lie in a comparable numeric range to mimic perceptual errors; and for *Operation*, they are plausible but incorrect within the same experimental setting. This design forces models to ground their answers in visual signals.

**Procedural Understanding.**   These four sequential tasks are built on step lists derived from ASR, The first is *Step Ordering*, where each segment's step sequence is converted into a four-option MCQ with distractors generated by DeepSeek-R1 as plausible but incorrect permutations that still follow experimental logic. The other three are formulated by embedding step list into question templates. *Sequence Generation* and *Step Prediction* use the full step list as the candidate set, where *Step Prediction*, additionally, the final step and its video are removed, with only segments containing at least three preceding steps retained; *Completeness Verification* instead uses the segment step list and randomly removes a non-final step as the target answer.

**Scientific Reasoning.** For *Experimental Analysis* and *Scientific Discovery*, we construct annotations for each full experiment video based on its corresponding peer-reviewed paper. The paper is first processed with MinerU (Wang et al., 2024a) to extract key sections (Introduction, Results, Discussion), and GPT-5 is used to summarize findings as anchors for annotation. PhD-level expert annotators then design two types of fill-in-the-blank question based on experiment videos and corresponding paper, under the following principles: 1) Solvable only through visual observation and requiring reasoning across the full experiment. 2) Should not be answerable without the video. 3) Constrained to a single precise answer, minimizing ambiguity and synonym overlap. 4) Should annotate multi-blank questions, where each question contains multiple blanks that capture several key information points.

**Expert Verification.** All annotations are rigorously human-verified through a structured expert workflow. For personnel capacity, we maintained a pool of roughly 15 domain-prepared annotators per major scientific category (e.g., medicine, biology), with a total team size of about 50 annotators, enabling balanced and timely distribution of tasks across disciplines.

To ensure consistent and scalable annotation across heterogeneous tasks, we built a dedicated online annotation platform (see illustrative examples in Appendix E) with task-specific interfaces tailored to each question type. These interfaces guide annotators through the rubric and enforce correct use of the annotation schema. Each annotation requires a brief justification, even for approvals, ensuring transparency and quality control.

Annotators follow unified criteria across all tasks:

- **Video-grounded:** All questions must be solvable using the visual evidence in the video.
- **No leakage or shortcuts:** Stems must not reveal the answer; distractors must be scientifically plausible.
- **Concrete, step-level fidelity:** Only visually verifiable actions are retained; abstract or unobservable descriptions are revised or removed.
- **Consistent formatting and clarity:** Wording avoids unverifiable details and ensures each question has a unique, unambiguous answer.
- **Justified verification:** Annotators provide reasons for all accepted or corrected items.

Each annotator first reviews the entire experiment video together with its associated paper (approximately 40 minutes per experiment), followed by question-level verification that averages 6–8 minutes for Level-1, 13 minutes for Level-2, and 18 minutes for Level-3.

The overall verification pipeline includes a one-month pilot phase for rubric alignment and iterative feedback, followed by one month of formal annotation, ultimately yielding 7,800 QA pairs across 10 tasks under 13 disciplines. Additional benchmark statistics are provided in Appendix B.2.

## 4 EXPERIMENTS

**Evaluation models.** We evaluate MLLMs covering both open-source and proprietary models, and reasoning ones or not. On the open-source side, we include Qwen2.5-VL (Bai et al., 2025c), Qwen3-VL (Bai et al., 2025b), InternVL3 (Zhu et al.), InternVL3.5 (Wang et al., 2025a), GLM4.5V (Hong et al., 2025), Kimi-VL (Team et al., 2025), and Intern-S1 (Bai et al., 2025a). For closed-source ones, we benchmark Seed-1.5-VL (Guo et al., 2025b), Gemini-2.5-Flash (DeepMind, 2025a), Gemini-2.5-Pro (DeepMind, 2025b), Claude-Sonnet-4 (Anthropic, 2025), and GPT-5 (OpenAI, 2025). A full description of the evaluated models' configurations can be found in Appendix F.

**Metrics.** ExpVid adopts hierarchical evaluation metrics aligned with task design.

- **Level-1:** All fine-grained recognition tasks are formulated as multiple-choice questions (MCQs). Performance is measured by Top-1 Accuracy, defined as the ratio of correctly answered questions to the total number of Level-1 questions.
- **Level-2.** This level comprises four task types: *Sequence Ordering*, *Completeness Verification*, *Step Prediction*, and *Sequence Generation*. The first three are MCQs. In *Sequence Ordering*, models select the correct step order from four candidates. In *Completeness Verification*, candidate options correspond to all steps within a specific video segment, resulting in instance-dependent

Table 2: Performance of evaluated models on the ExpVid across 10 tasks under three levels.

| Model | Think | Level-1 | | | | | Level-2 | | | | | Level-3 | | |
|---|---|---|---|---|---|---|---|---|---|---|---|---|---|---|
| | | Tool | Mat. | Quan. | Oper. | Avg. | Ord. | Gen. | Veri. | Pred. | Avg. | Anal. | Disc. | Avg. |
| Human Performance | | 17.5 | 15.9 | 61.3 | 55.5 | 37.6 | 69.8 | 31.2 | 45.6 | 21.8 | 42.1 | – | – | – |
| *Open-source MLLMs* | | | | | | | | | | | | | | |
| Qwen2.5-VL-7B-Instruct | × | 32.0 | 33.9 | 49.0 | 62.4 | 42.6 | 56.2 | 20.8 | 20.7 | 1.3 | 24.6 | 25.2 | 21.4 | 23.3 |
| MiMo-VL-7B-RL | × | 34.2 | 33.7 | 44.2 | 62.4 | 42.4 | 43.9 | 28.5 | 18.5 | 11.4 | 27.4 | 28.7 | 25.9 | 27.3 |
| MiMo-VL-7B-RL | ✓ | 36.1 | 29.1 | 53.6 | 67.8 | 44.3 | 64.8 | 32.3 | 24.9 | 15.6 | 34.3 | 29.3 | 27.3 | 28.3 |
| InternVL3-8B | × | 27.5 | 31.0 | 38.8 | 65.6 | 39.4 | 43.4 | 20.4 | 20.2 | 3.9 | 23.9 | 29.2 | 25.3 | 27.2 |
| InternVL3.5-8B | × | 27.3 | 30.8 | 45.5 | 64.8 | 40.3 | 82.3 | 25.8 | 23.7 | 4.8 | 34.0 | 22.6 | 18.4 | 20.5 |
| Intern-S1-mini | ✓ | 33.3 | 31.2 | 52.5 | 61.4 | 42.5 | 73.6 | 14.3 | 16.8 | 8.3 | 28.1 | 33.5 | 28.3 | 30.9 |
| Keye-VL-8B-Preview | ✓ | 16.6 | 22.4 | 38.9 | 60.8 | 32.6 | 25.4 | 12.4 | 19.1 | 1.7 | 14.6 | 9.5 | 6.7 | 8.1 |
| Keye-VL-1.5-8B | ✓ | 21.0 | 23.4 | 51.3 | 64.0 | 37.0 | 56.7 | 9.5 | 20.0 | 2.8 | 22.1 | 8.4 | 6.1 | 7.2 |
| GLM-4.1V-9B | ✓ | 30.8 | 29.8 | 47.5 | 59.6 | 40.1 | 64.1 | 18.2 | 25.0 | 7.4 | 28.6 | 28.1 | 26.5 | 27.3 |
| GLM-4.5V | ✓ | 35.5 | 33.6 | 61.5 | 62.3 | 45.6 | 71.9 | 34.9 | 27.2 | 12.9 | 36.6 | 33.3 | 32.5 | 32.9 |
| Kimi-VL-A3B-Thinking | ✓ | 34.6 | 32.6 | 40.7 | 59.5 | 40.8 | 32.3 | 18.2 | 23.3 | 6.2 | 20.0 | 24.6 | 21.8 | 23.2 |
| InternVL3.5-38B | ✓ | 35.9 | 34.0 | 46.7 | 65.3 | 44.0 | 65.8 | 36.7 | 23.0 | 19.0 | 36.0 | 33.1 | 30.8 | 31.9 |
| InternVL3-78B | ✓ | 35.1 | 34.3 | **73.2** | 75.8 | 50.9 | **87.1** | 45.5 | 19.8 | 15.5 | 41.9 | 40.3 | 35.3 | 37.7 |
| Qwen2.5-VL-72B-Instruct | × | 30.5 | 34.7 | 54.5 | 64.5 | 43.9 | 86.3 | 34.1 | 23.8 | 0.3 | 35.9 | 31.9 | 29.3 | 30.6 |
| Qwen3-VL-235B-Instruct | × | 30.2 | 34.1 | 50.0 | 68.5 | 45.1 | 74.0 | 42.3 | 16.2 | 19.6 | 39.0 | 40.5 | 31.8 | 36.1 |
| Qwen3-VL-235B-Think-L | ✓ | 35.7 | 30.2 | 48.6 | 68.8 | 43.9 | 69.9 | 36.6 | 20.3 | 24.3 | 37.8 | 30.2 | 36.3 | 33.1 |
| Qwen3-VL-235B-Think-H | ✓ | 36.6 | 31.0 | 48.6 | 71.0 | 44.9 | 68.5 | 38.3 | 21.6 | 27.0 | 38.9 | 32.2 | 38.1 | 35.1 |
| Intern-S1 | ✓ | 38.9 | 35.2 | 58.9 | 73.8 | 49.9 | 82.2 | 45.0 | 24.1 | 15.4 | 36.0 | 43.0 | 36.3 | 39.6 |
| *Closed-source MLLMs* | | | | | | | | | | | | | | |
| Seed-VL-1.5 | ✓ | 32.9 | 24.6 | 43.9 | 69.2 | 40.7 | 73.9 | 48.6 | 19.8 | 27.9 | 42.5 | 32.0 | 29.4 | 30.7 |
| Claude-Sonnet-4 | × | 25.6 | 31.2 | 54.3 | 61.9 | 40.8 | 78.7 | 37.6 | 16.5 | 11.6 | 36.0 | 29.1 | 30.1 | 29.6 |
| Gemini-2.5-Flash | × | 52.7 | 50.1 | 65.2 | 72.6 | 58.6 | 86.0 | 50.5 | 24.1 | 40.2 | 50.1 | 47.2 | 41.1 | 44.1 |
| Gemini-2.5-Flash | ✓ | 52.7 | **50.7** | 71.9 | 73.3 | **60.2** | 85.1 | 54.3 | 22.3 | 38.0 | 49.8 | 44.8 | 41.3 | 43.0 |
| Gemini-2.5-Pro | × | **53.1** | 45.9 | 64.3 | **80.8** | 59.2 | 83.7 | 61.3 | **26.8** | 49.6 | 53.8 | 50.6 | 45.2 | 47.9 |
| Gemini-2.5-Pro | ✓ | 51.3 | 44.3 | 63.8 | 74.4 | 56.7 | 84.2 | 59.9 | **26.8** | 46.9 | 54.3 | 50.1 | 44.8 | 47.4 |
| GPT-5 | ✓ | 51.6 | 37.8 | 59.5 | 71.9 | 53.3 | 85.1 | **66.9** | **26.8** | 51.8 | **57.5** | 55.4 | 57.4 | **56.4** |

option counts (see Fig. 14, left). In *Step Prediction*, options are drawn from all steps in the full experimental procedure, also leading to variable option sizes (see Fig. 14, right). These MCQ tasks are evaluated using Top-1 Accuracy.

*Sequence Generation* requires producing an ordered sequence of steps and is evaluated using the Jaccard Similarity Coefficient (range: 0–1), measuring overlap between the predicted and ground-truth step sets. The overall Level-2 score is computed by averaging Top-1 Accuracy for MCQ tasks and similarity scores for Sequence Generation across all Level-2 instances.

• **Level-3.** This level contains two tasks: *Experimental Analysis* and *Scientific Discovery*. All questions are formulated as fill-in-the-blank. We employ a lightweight language model to compare model outputs with reference answers. Each blank is assigned one point, and performance is measured by Per-blank Accuracy, defined as the number of correctly filled blanks divided by the total number of blanks.

**Human performance.** We recruited 15 undergraduate students without specialized backgrounds in biomedical or related sciences. They represent participants with general knowledge and common sense rather than domain expertise, providing a realistic reference point for non-expert human understanding. Notably, for Level-3 open-ended cloze tasks, participants reported being unable to complete the questions without specialized training, so no human baseline is reported for this level.

## 4.1 RESULTS

We evaluate 20 MLLMs on ExpVid, as detailed in Tab. 2. Frontier closed-source models, notably GPT-5 and the Gemini-2.5 series, clearly outperform the human baseline. Gemini-2.5-Flash-Think

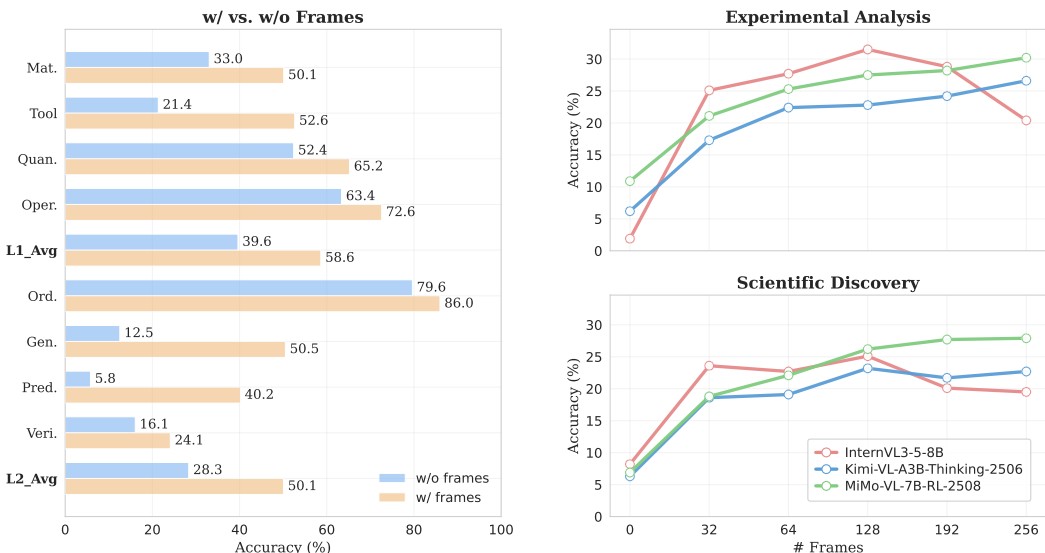

Figure 3: Effect of input video frames.

reaches 60.2 on the Level-1 (L1) average, and GPT-5 scores 57.5 on the Level-2 (L2) average, well above the human averages of 37.6 and 42.1, respectively.

Closed-source models also maintain a clear lead over open-source ones as shown in Tab. 2, a gap that widens with task complexity. In basic perception such as recognizing tools, materials, quantities, and operations, closed-source models hold a notable lead. The top-performing Gemini-2.5-Flash (with "think") scores 60.2 on average. The best open-source models, InternVL3-78B and Intern-S1, achieve commendable but lower scores of 50.9 and 49.9, respectively. This indicates that while the gap exists, leading open-source models are becoming increasingly competitive in fundamental visual perception.

Concerning procedural understanding, the gap becomes more pronounced. GPT-5 leads with an average of 57.5, followed closely by Gemini-2.5-Pro at 54.3. The top open-source model, InternVL3-78B, lags with an average of 41.9. A deeper look reveals nuances: InternVL3-78B excels at Step Ordering (87.1), even outperforming GPT-5 (85.1). However, it falls short on more generative and predictive tasks like Sequence Generation (45.5 vs. GPT-5's 66.9) and Step Prediction (15.5 vs. GPT-5's 51.8). This highlights that while open-source models can master specific structured tasks, they struggle with more holistic procedural reasoning.

In Level-3 (L3) scientific reasoning, GPT-5 achieves a leading average score of 56.4, with strong results in both Experimental Analysis (55.4) and Scientific Discovery (57.4), well ahead of all competitors. By contrast, the best open-source model, Intern-S1, reaches only 39.6, falling nearly 17 points short of GPT-5. It underscores the advanced reasoning capabilities of frontier closed-source models, which remain a clear target for the open-source community.

## 4.2 MORE ANALYSIS

**Scaling Effects in Open-Source Models.** A clear and consistent trend found among open-source models is the positive correlation between model scale and performance. The InternVL family serves as an excellent case study. As the model size increases from InternVL3-8B (L1: 39.4, L2: 23.9, L3: 27.2) to InternVL3.5-38B (L1: 44.0, L2: 36.0, L3: 31.9) and finally to InternVL3-78B (L1: 50.9, L2: 41.9, L3: 37.7), performance improves across all three levels. This demonstrates that increasing model scale directly contributes to enhanced capabilities in perception, procedural understanding, and scientific reasoning tasks, validating scaling as a crucial axis for experiment video understanding in the open-source ecosystem.

**Potential Unbalanced Capabilities.** The results also shed light on the relative difficulty of different tasks. Within L2, models consistently score highest on Step Ordering, indicating a strong

ability to rearrange provided information. In contrast, scores for Completeness Verification and Step Prediction are significantly lower across all models, revealing a weakness in identifying missing information and forecasting future actions. The extremely low score of Qwen2.5-VL-72B-Instruct on Step Prediction (0.3) despite its strong performance on Step Ordering (86.3) exemplifies the brittleness and uneven capabilities of current MLLMs.

**Effect of thinking.** In Tab. 2, we find *Thinking* does not consistently improve results. The patterns from Qwen3-VL under different thinking budgets further show that a high thinking budget can even reduce accuracy on several tasks. Regarding this, we analyze error cases where Gemini-2.5-Flash with Thinking_Budget=8,192 fails but the NoThinking mode succeeds. The Thinking model often adopts a logic-oriented style: abstracting the problem, reasoning step by step, and proposing a "reasonable" workflow. Yet it drifts from the actual video sequence and relies on priors. By contrast, the NoThinking model remains video-grounded, directly matching steps to visual order and producing concise, faithful descriptions. For example, NoThinking answers typically begin with *"The video shows..."*, whereas Thinking answers start with *"...identify the most logical workflow..."*, revealing reasoning beyond visuals (see Appendix F.4).

**Vision centric.** We compare Gemini-2.5-Flash with and without frame inputs on all L1 and L2 tasks (the left of Fig. 3). As a result, inputting frames consistently boosts performance, with some tasks such as Step Prediction becoming unsolvable without visual cues. Even for tasks like Step Ordering, where models can sometimes infer the correct answer from scientific priors alone, adding video inputs still yields clear gains. This validates the vision-centric design of ExpVid.

For long-video reasoning tasks in L3, we ablate frame counts in Fig. 3 right. Results show that visuals are indispensable: accuracy is near zero without frames and increases as more are added. However, models benefit differently. InternVL3.5 peaks early ($\sim$128 frames) and then declines, suggesting saturation or distraction from redundant inputs, whereas MiMo-VL and Kimi-VL steadily improve up to 256 frames, reflecting stronger ability to leverage extended temporal context. This indicates MLLMs like InternVL3.5, trained mainly for image–text alignment, gain little from extended sequences. In contrast, Kimi-VL and MiMo-VL, which incorporated long-video data during long-context activation training, continue to improve with more frames. Overall, these findings highlight the critical role of vision and the varying optimal frame budgets across models.

**Limitation.** ExpVid currently focuses on wet-lab experiments, not covering the full spectrum of scientific inquiry. Domains such as physics, which often involve distinct experimental apparatus (e.g., optical tables, particle detectors) and abstract phenomena, or purely computational experiments and large-scale engineering tests, remain underexplored. Reasoning tasks in Level-3 assess outcomes but do not illuminate the underlying reasoning process (e.g., chain-of-thought) that links experiments to conclusions.

## 5 CONCLUSION

This paper presents ExpVid, the first benchmark dedicated to scientific experiment videos. With its three-level task hierarchy, vision-centric annotation pipeline, and expert-guided validation, ExpVid gives a systematic evaluation of MLLMs across fine-grained perception, procedural understanding, and scientific reasoning. Our empirical studies demonstrates both the progress and the persistent limitations of current models, highlighting directions for advancing trustworthy AI in experimental science.

## ACKNOWLEDGMENTS

We thank Shanghai Artificial Intelligence Laboratory for institutional support. We also sincerely acknowledge the domain expert annotators for their careful annotation and verification efforts in constructing the benchmark.

ETHICS STATEMENT

Our work involves the collection and annotation of scientific experiment videos sourced from JoVE, a peer-reviewed video journal. All data are publicly available under JoVE's license, and we do not involve any private, sensitive, or personally identifiable information. The benchmark focuses on laboratory procedures rather than human subjects, and no clinical or personally invasive data are included. Annotation was conducted by PhD-level domain experts with clear guidelines to ensure accuracy, fairness, and scientific integrity. Potential risks such as misuse for non-scientific or unsafe experimental replication are mitigated by providing the dataset strictly for research purposes. We adhere to the ICLR Code of Ethics in all aspects of this work, including dataset release, annotation transparency, and reporting of model limitations.

REPRODUCIBILITY STATEMENT

We have taken several steps to ensure reproducibility of our benchmark and experiments. Sec. 3.1 and Appendix B.1 describe data collection and filtering criteria, including quantitative thresholds. Sec. 3.1 details preprocessing pipelines for constructing our benchmark. Sec. 3.3, Appendix E and G outline annotation templates, distractor generation heuristics, and expert verification processes. Evaluation protocols and metrics for all tasks are specified in Sec. 4 and Appendix F.

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

# A  THE USAGE OF LARGE LANGUAGE MODELS (LLMs)

In our work, LLMs are employed to assist the automated data annotation pipeline, with the resulting annotations subsequently reviewed and refined by human researchers. In addition, LLMs are used to support proofreading of the manuscript. All content presented in this paper is rigorously verified to ensure faithful representation of the authors' original intent and to eliminate any factual inaccuracies or hallucinations that might be introduced by the models.

# B  DATASET STATISTICS

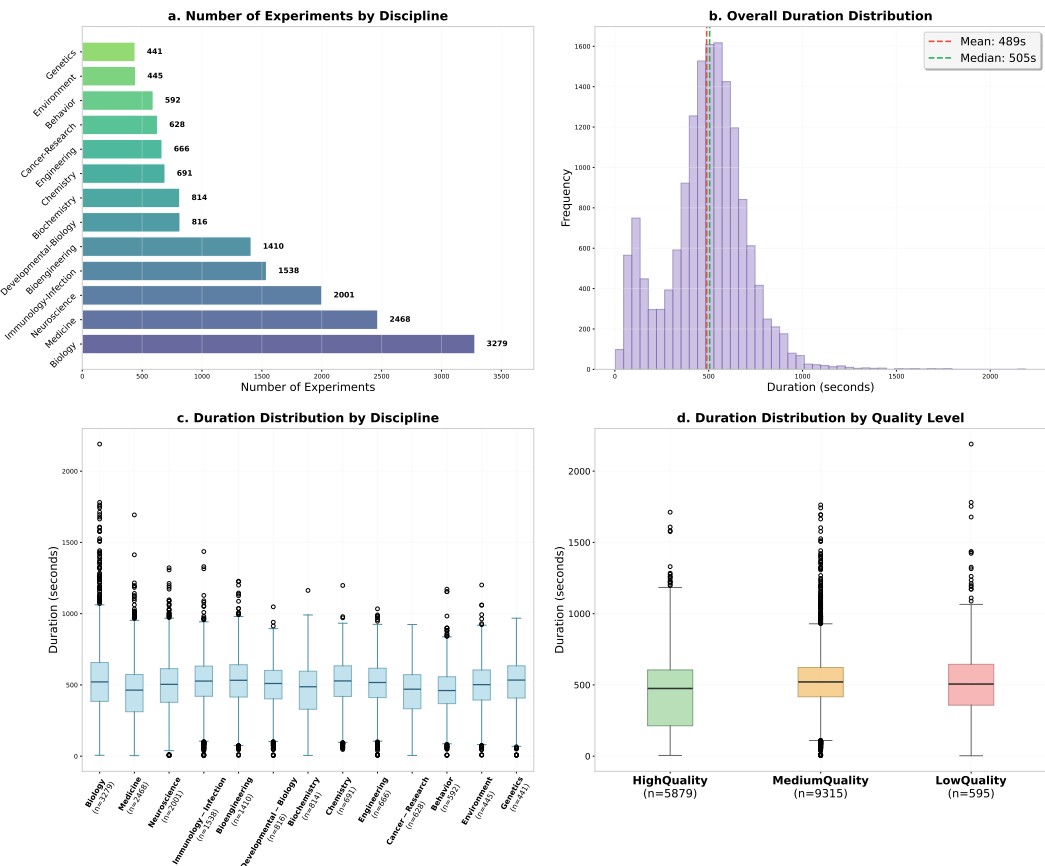

Figure 4: Data statistics in ExpVid collection and filtering. (a) Number of experiment videos per discipline before filtering. (b) Video duration distribution with mean 489s and median 505s, showing long-tail outliers beyond 2,000s. (c) Boxplot of video duration by discipline (whiskers at 1.5×IQR). (d) Boxplot of video duration by quality based on the multi-dimensional scoring process.

In this section, we present key statistics of ExpVid and its curation process.

## B.1  STATISTICS IN DATA COLLECTION AND FILTERING

Fig. 4 shows the overall video duration distribution, the number of experiments across disciplines, and the results of the multi-dimensional scoring process. As illustrated, the source collection (JoVE) initially contains tens of thousands of videos, with biology, medicine, and neuroscience among the largest disciplines. The raw duration distribution centers around 489s on average (median 505s), but includes long-tail outliers exceeding 2,000s.

To ensure high quality, we retain only experiments with an overall score of at least 4 and no individual dimension score below 3.5, resulting in 5,879 videos (37.2%). To further align with our

task hierarchy and maintain temporal diversity, we exclude videos outside the interquartile range (378s–728s). After this coarse filtering guided by LLM-based ASR scoring, a multidisciplinary expert team manually curated the final dataset. To balance disciplines, control annotation cost, and keep a manageable benchmark size, we preserve 30 experiments per discipline across 13 fields, yielding 390 experiments in total.

The 13 disciplines include: Genetics, Environment, Behavior, Cancer Research, Engineering, Chemistry, Biochemistry, Developmental Biology, Bioengineering, Immunology and Infection, Neuroscience, Medicine, and Biology.

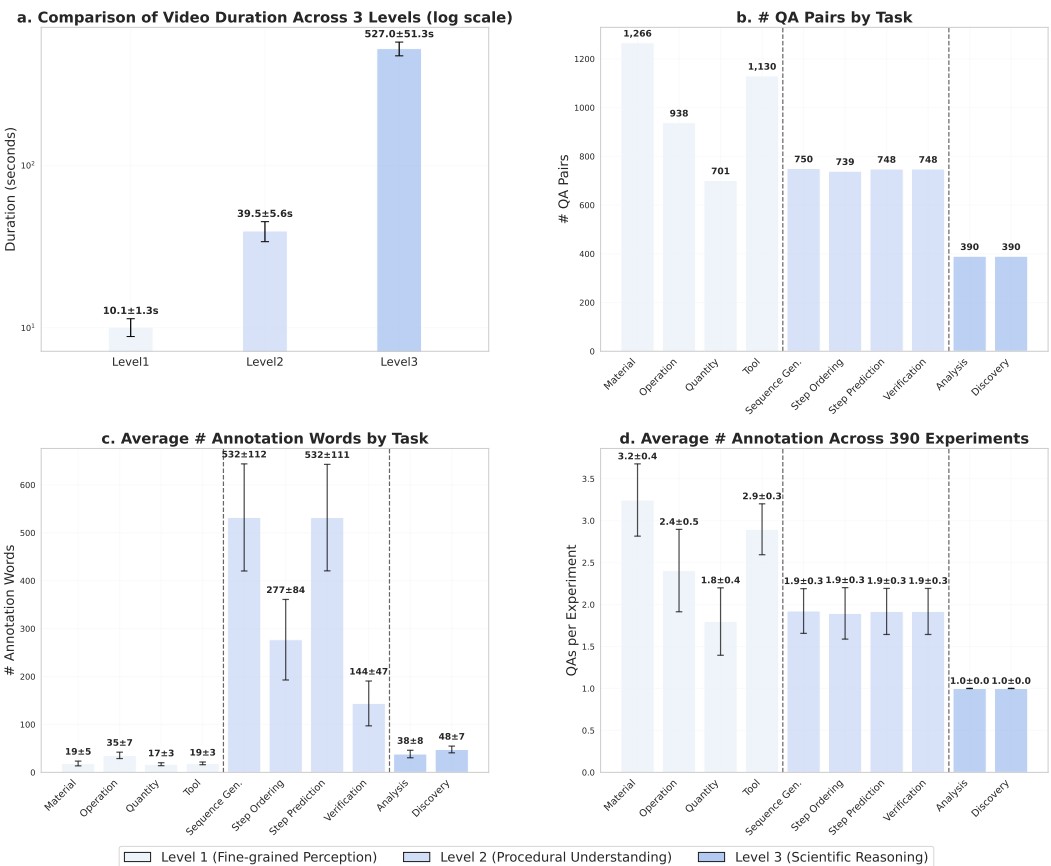

Figure 5: Data statistics of video duration and annotations in ExpVid. (a) Average video/clip duration and standard deviation across the three levels (log scale). (b) Number of annotations for each task. (c) Average number of words per annotation with standard deviation. (d) Average number of annotations per full experimental video across different tasks, with standard deviation.

## B.2 STATISTICS IN CURATED BENCHMARK

We further provide detailed statistics of the annotated dataset in Fig. 5. As shown in Fig. 5 (a), our preprocessing splits videos into three levels with relatively stable durations and small standard deviations. In particular, the small variance at Level-3 benefits from the filtering process, which controls video length during selection. The progressively longer durations across the three levels naturally support our design for evaluating different capabilities, emphasizing not only linguistic reasoning but also reasoning across temporal scales.

Fig. 5 (c) reports the token counts of annotated tasks. Sequence generation and step prediction at Level-2 contain significantly more tokens than other tasks, since their questions include the predefined full step list as context. This indicates that models must reason over multi-step procedures in video while simultaneously handling long textual contexts.

Fig. 5 (d) shows the number of annotations per experiment across disciplines. Since we balance the number of experiments per discipline in filtering, the small variance here reflects that ExpVid spans diverse domains while maintaining annotation consistency, ensuring fair evaluation of models' cross-disciplinary capabilities.

## C PERFORMANCE BY DISCIPLINE

We visualize the averaged performance on each task by discipline in Fig. 6. The figure shows that, because these disciplines are closely related and primarily consist of web-based experiments, the performance differences across disciplines remain limited.

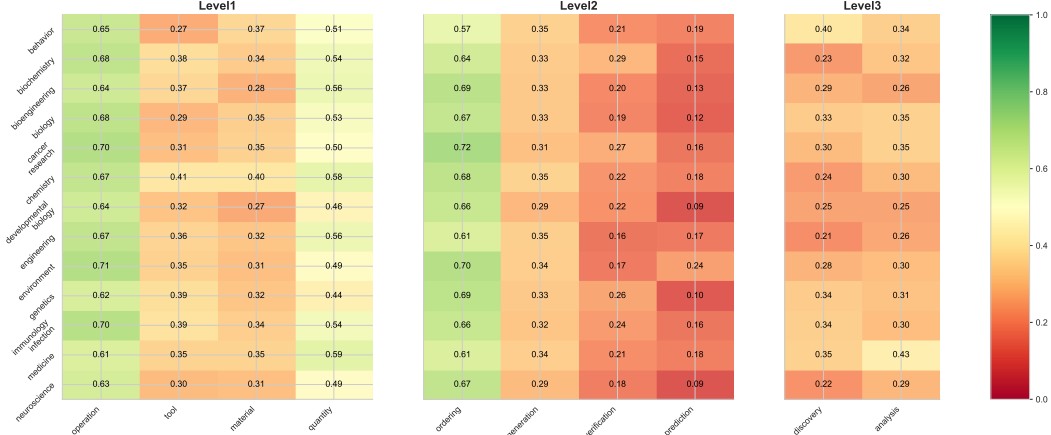

Figure 6: Three level performance averaged across models by disciplines.

## D   EXAMPLES OF EACH TASK

In this section, we provide representative QA examples for every task across all three levels.

For Level-1, we include video frame snapshots to facilitate clearer visual interpretation, as shown in Fig. 7. In particular, for the Quantity Recognition task, we provide multiple examples accompanied by the human annotators' accepted reasoning and supporting evidence from the corresponding video frames, illustrating how the task is grounded in fine-grained visual cues.

For Level-2 and Level-3, due to the substantially longer temporal scale of the videos, we present key frames for each QA example in Figs. 8, 9, and 10. The complete video examples can be accessed through the anonymous link provided above.

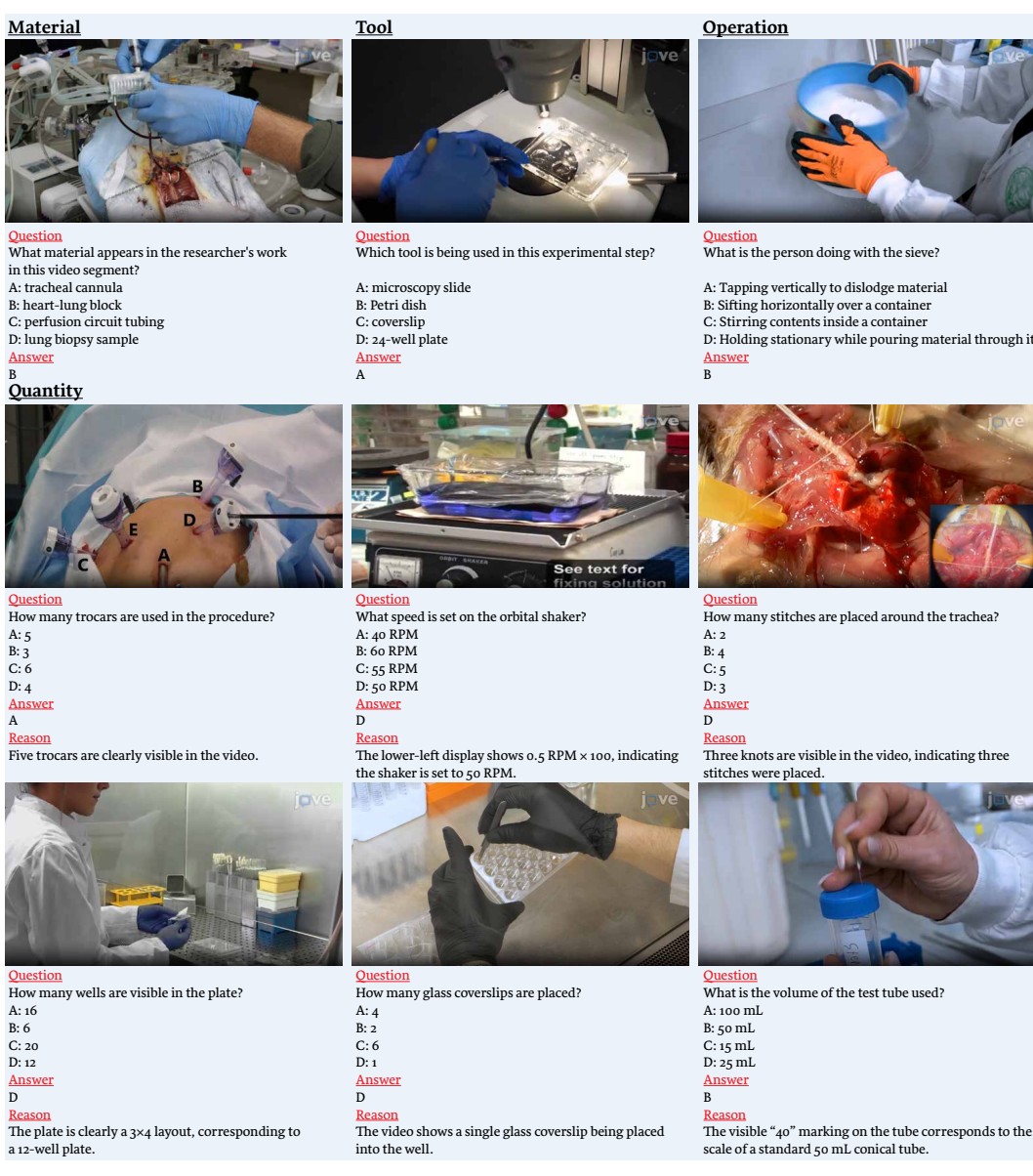

Figure 7: Level-1 QA examples, including Material, Tool, Operation and Quantity Recognition.

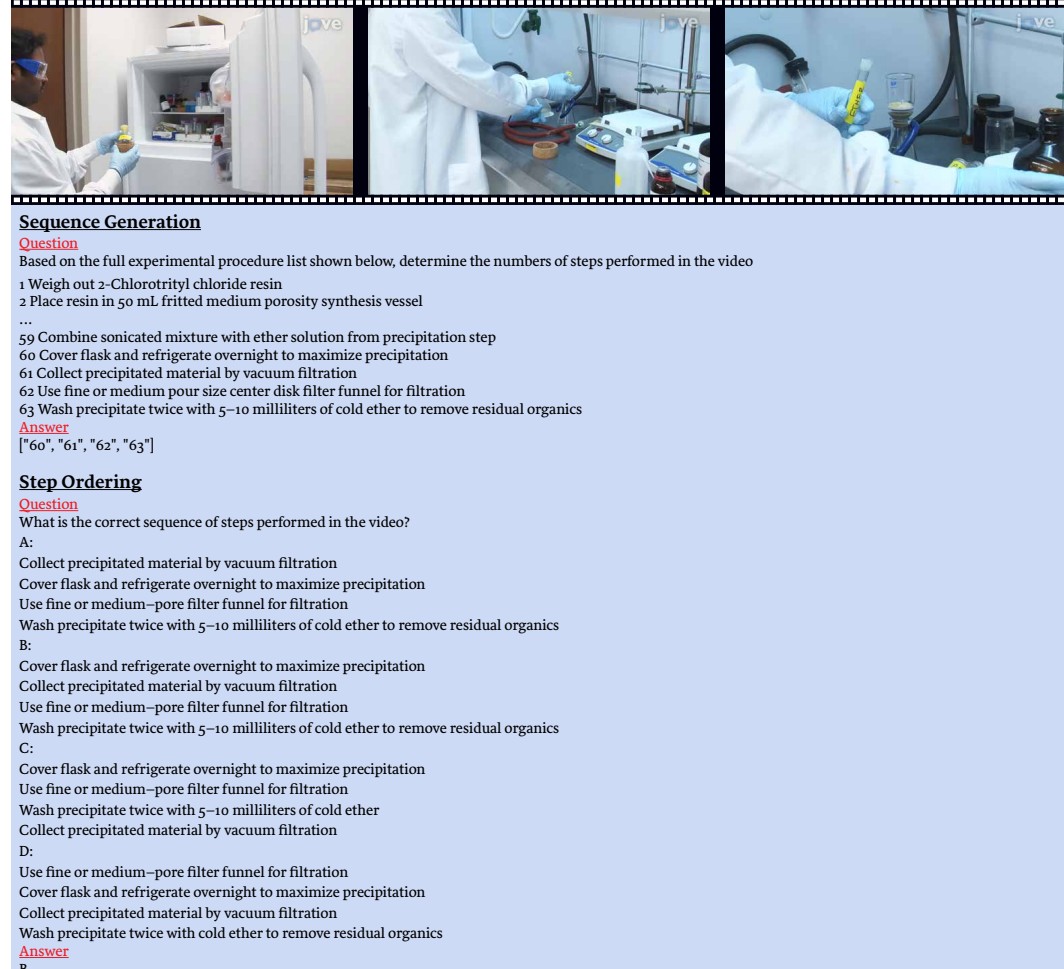

**Sequence Generation**

Question

Based on the full experimental procedure list shown below, determine the numbers of steps performed in the video

1 Weigh out 2-Chlorotrityl chloride resin
2 Place resin in 50 mL fritted medium porosity synthesis vessel
...
59 Combine sonicated mixture with ether solution from precipitation step
60 Cover flask and refrigerate overnight to maximize precipitation
61 Collect precipitated material by vacuum filtration
62 Use fine or medium pour size center disk filter funnel for filtration
63 Wash precipitate twice with 5–10 milliliters of cold ether to remove residual organics

Answer

["60", "61", "62", "63"]

**Step Ordering**

Question

What is the correct sequence of steps performed in the video?

A:
Collect precipitated material by vacuum filtration
Cover flask and refrigerate overnight to maximize precipitation
Use fine or medium–pore filter funnel for filtration
Wash precipitate twice with 5–10 milliliters of cold ether to remove residual organics

B:
Cover flask and refrigerate overnight to maximize precipitation
Collect precipitated material by vacuum filtration
Use fine or medium–pore filter funnel for filtration
Wash precipitate twice with 5–10 milliliters of cold ether to remove residual organics

C:
Cover flask and refrigerate overnight to maximize precipitation
Use fine or medium–pore filter funnel for filtration
Wash precipitate twice with 5–10 milliliters of cold ether
Collect precipitated material by vacuum filtration

D:
Use fine or medium–pore filter funnel for filtration
Cover flask and refrigerate overnight to maximize precipitation
Collect precipitated material by vacuum filtration
Wash precipitate twice with cold ether to remove residual organics

Answer

B

Figure 8: Level-2 QA examples, including Step Ordering and Sequence Generation.

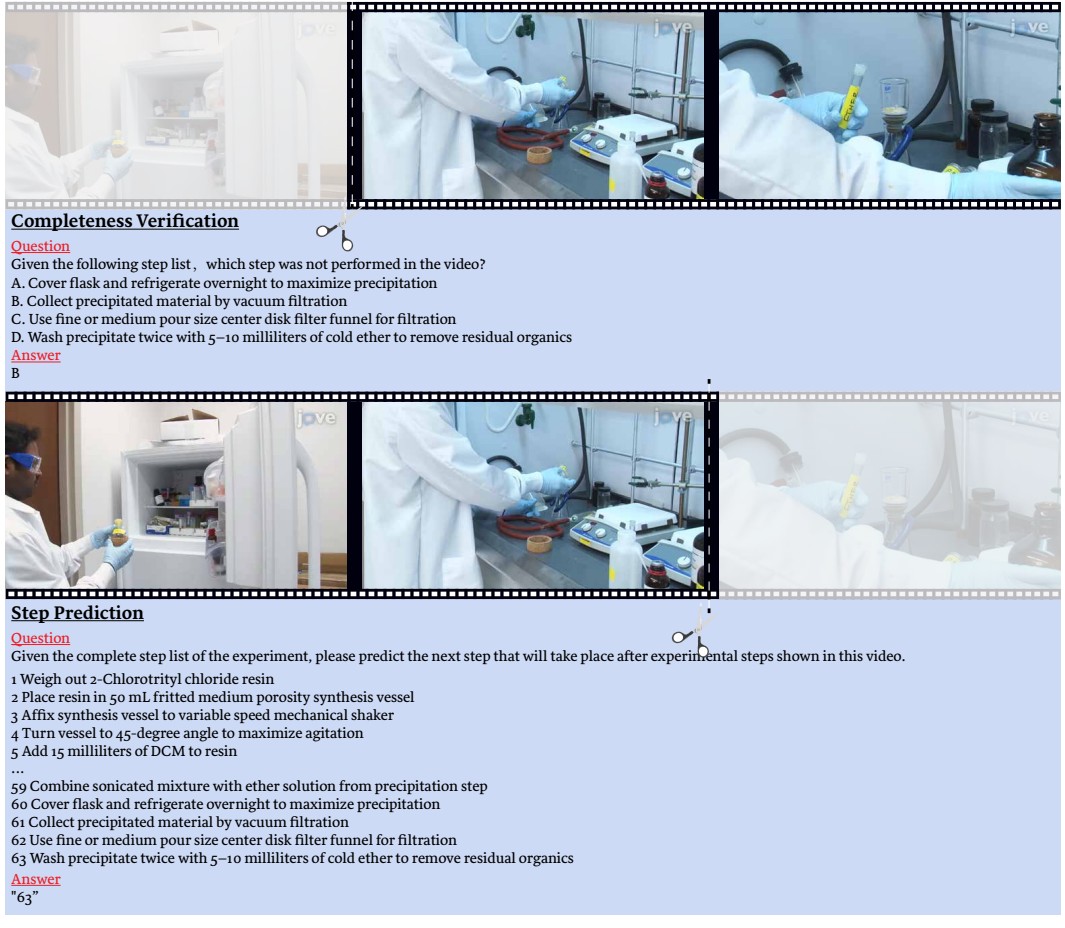

Figure 9: Level-2 QA examples, including Completeness Verification and Step Prediction. Target steps are clipped in the give video segments.

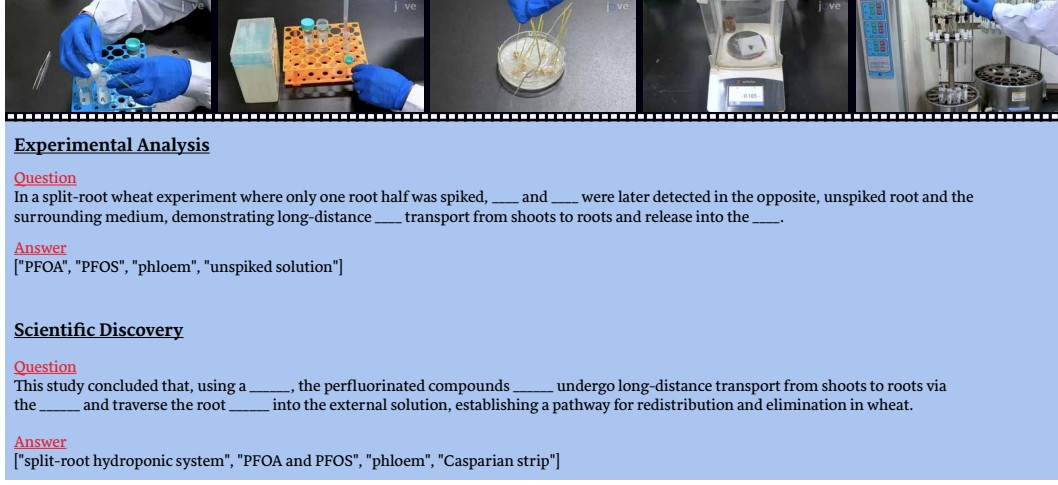

Figure 10: Level-3 QA examples, including Experimental Analysis and Scientific Discovery fill-in-blanks.

# E    EXPERT VERIFICATION

In this section, we present illustrative examples of the online annotation platform that supports expert verification across all tasks. Experts follow standardized guidelines: watch source videos and related materials, review annotations, and refine them to meet task-specific criteria. For any modifications, they must also provide justifications to ensure transparency and traceability.

Figs. 11, 12, and 13 show representative cases from each level. Experts validate annotations, correct errors, and refine question–answer pairs to ensure accuracy and domain fidelity. Level-3 is distinct in requiring annotators to also consult the corresponding research papers when designing questions. The entire process is iterative: low-quality annotations can be returned for revision until they fully satisfy the benchmark's standards.

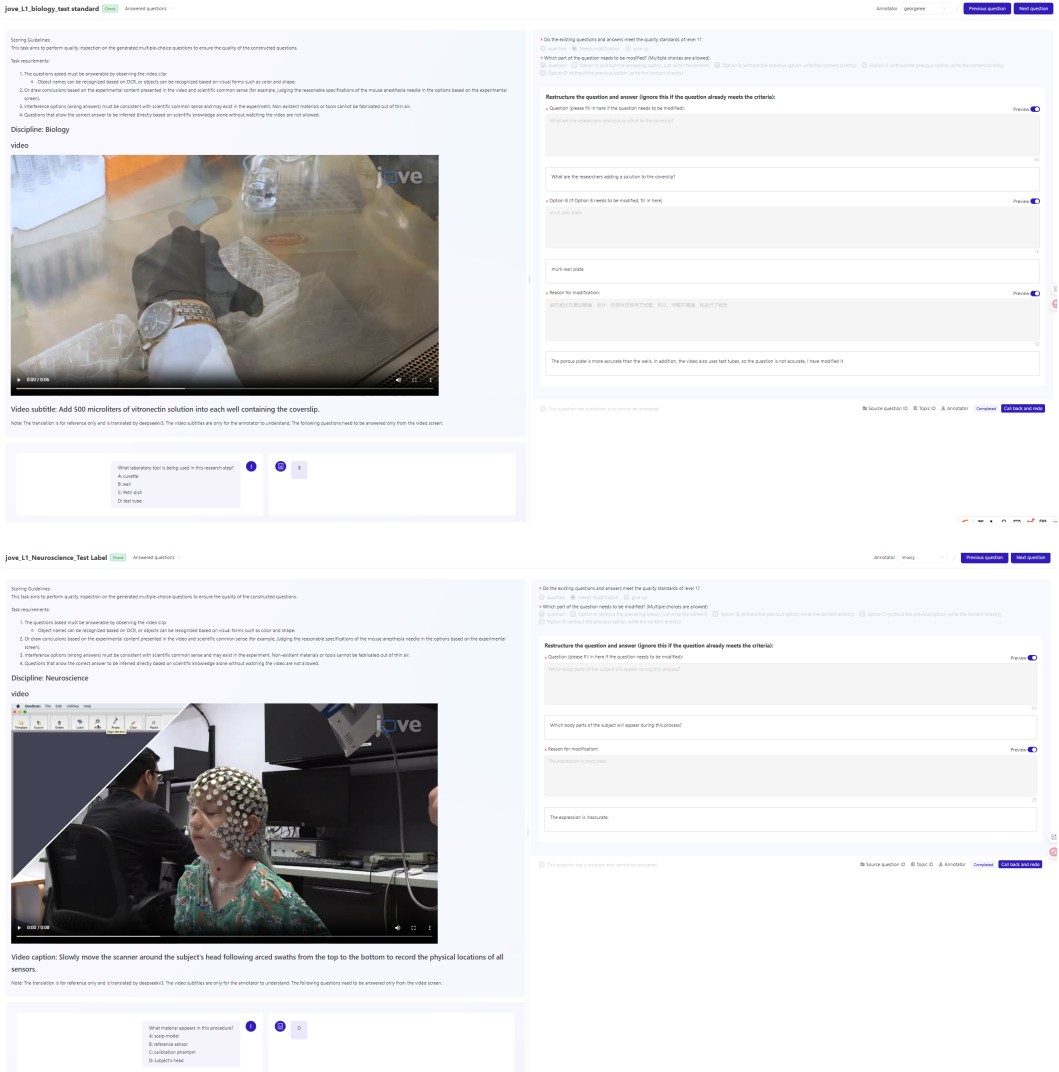

Figure 11: Expert annotation example of Level-1 task.

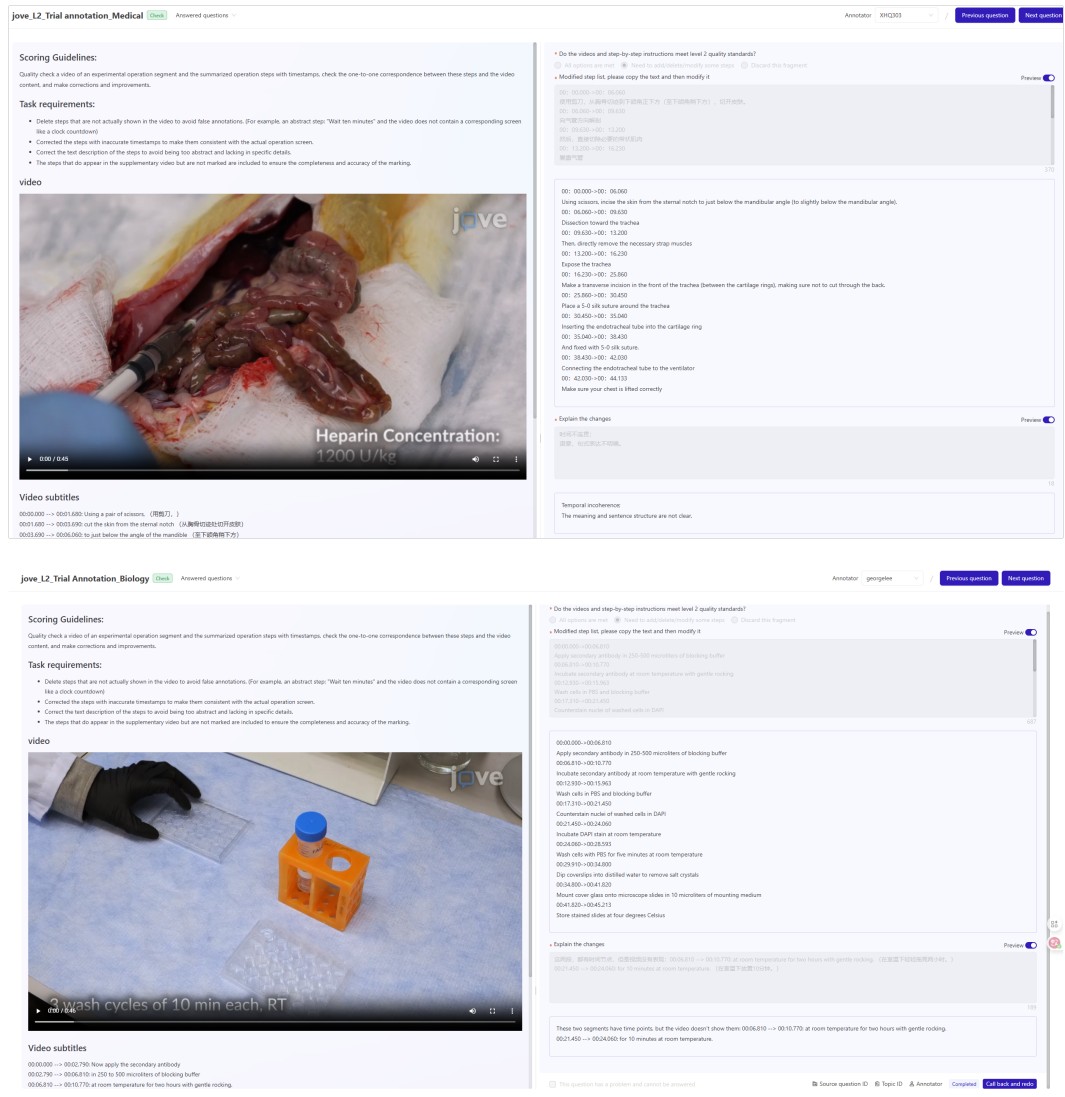

Figure 12: Expert annotation example of Level-2 task.

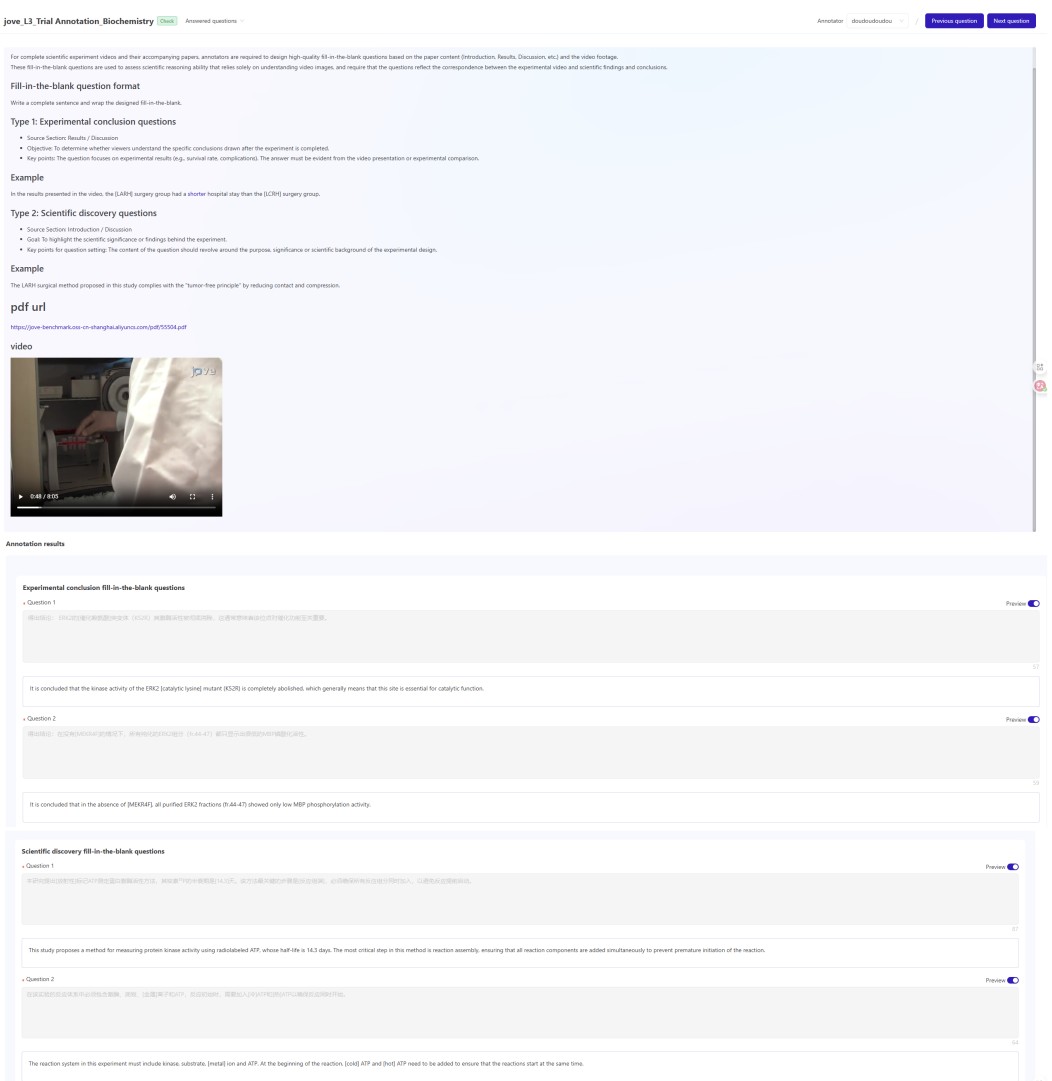

Figure 13: Expert annotation example of Level-3 task.

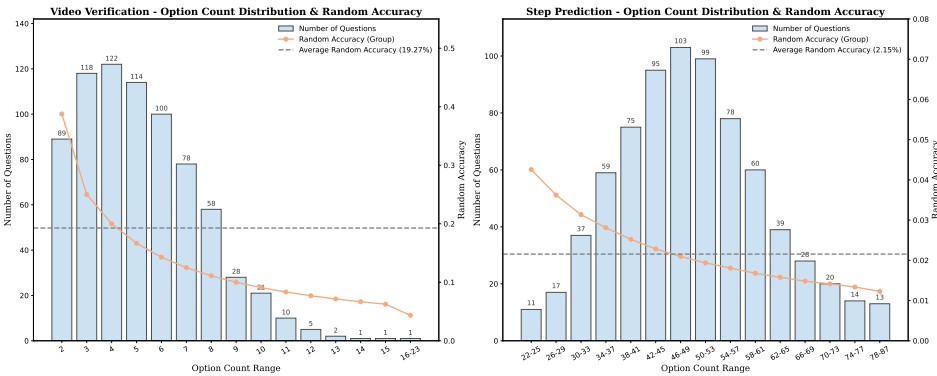

Figure 14: Distribution of # options of Completion Verification and Step Prediction.

# F EVALUATION DETAILS

In this section, we detail experiment settings, model configurations and inference prompts below.

## F.1 EXPERIMENT SETTINGS

For frame selection, we use 8 frames for Level 1 tasks and 32 frames for Level 2 tasks, which approximately correspond to a sampling rate of 1 fps given the average duration of the videos in these tasks. For Level 3 tasks, we adopt either the recommended number of frames or the maximum number of frames that can be accommodated within the model's context window and available GPU memory. Frames are uniformly sampled from the raw videos and resized to 224×224 to ensure fair comparison across models.

For inference, we allocate a maximum of 8192 tokens to each model to ensure that complete answers can be generated in the vast majority of instances. The temperature is fixed at 0.1 for all models to reduce randomness in generation.

## F.2 Configurations of Evaluated Models

The detailed configurations of evaluated MLLMs, including model versions and visual frame inputs, are given in Tab. 3.

Table 3: Details of evaluated MLLMs used in ExpVid. The "# Frames" column represents the default number of input frames in level3 tasks, chosen from {96, 128, 256, 512}. "HF" means Hugging Face inference, "vLLM" indicates vLLM engine, and "API" denotes proprietary API call.

| Organization | Model | Release | Version | Level3 # Frames | Pipeline |
|---|---|---|---|---|---|
| *Closed-source MLLMs* | | | | | |
| OpenAI | GPT-5 | 2025-8 | `GPT-5` | 128 | API |
| Google | Gemini-2.5-Flash | 2025-5 | `Gemini-2.5-Flash` | 128 | API |
| | Gemini-2.5-Pro | 2025-3 | `Gemini-2.5-Pro` | 128 | API |
| Anthropic | Claude-Sonnet-4 | 2025-5 | `Claude-Sonnet-4` | 96 | API |
| ByteDance | Seed1.5-VL | 2025-5 | `Seed1.5-VL` | 256 | API |
| *Open-source MLLMs* | | | | | |
| Alibaba | Qwen2.5-VL-7B | 2025-1 | `Qwen2.5-VL-7B-Instruct` | 128 | vLLM |
| | Qwen2.5-VL-72B | 2025-1 | `Qwen2.5-VL-72B-Instruct` | 128 | vLLM |
| | Qwen3-VL-235B | 2025-9 | `Qwen3-VL-235B-A22B-Instruct` | 128 | vLLM |
| | Qwen3-VL-235B | 2025-9 | `Qwen3-VL-235B-A22B-Thinking` | 128 | vLLM |
| Shanghai AI Lab | InternVL3-8B | 2025-4 | `InternVL3-8B` | 256 | HF |
| | InternVL3.5-8B | 2025-9 | `InternVL3.5-8B` | 256 | HF |
| | InternVL3.5-38B | 2025-9 | `InternVL3.5-38B` | 256 | HF |
| | InternVL3-78B | 2025-4 | `InternVL3-78B` | 256 | HF |
| | Intern-S1-mini | 2025-7 | `Intern-S1-mini` | 128 | HF |
| | Intern-S1 | 2025-7 | `Intern-S1` | 128 | HF |
| Kwai | Keye-VL-8B-Preview | 2025-6 | `Keye-VL-8B-Preview` | 256 | HF |
| | Keye-VL-1.5-8B | 2025-9 | `Keye-VL-1.5-8B` | 256 | HF |
| Moonshot | Kimi-VL-A3B-Thinking | 2025-6 | `Kimi-VL-A3B-Thinking-2506` | 256 | vLLM |
| Xiaomi | MiMo-VL-7B-RL | 2025-8 | `MiMo-VL-7B-RL-2508` | 512 | vLLM |
| ZhipuAI | GLM-4.1V-9B-Thinking | 2025-7 | `GLM-4.1V-9B-Thinking` | 256 | HF |
| | GLM-4.5V | 2025-8 | `GLM-4.5V` | 256 | API |

### F.3 PROMPT FOR INFERENCE

We provide prompt templates across all tasks and examples below.

---

**Prompt for Level 1 Tasks**

**Full Prompt**
{task_instruction}
{question}

**Task Instruction**
Solve the multiple choice question based on the video. Provide your final answer as a single letter enclosed in \boxed{}.

**Question**

**Materials**
Question: Which material appears in this experimental step?
Options:
A: collected pellets
B: agarose beads
C: silica gel packets
D: lyophilized powder

**Operation**
Question: What is the person doing with the pipette to the cell plate wells?
Options:
A: Removing the medium
B: Pouring fresh medium
C: Injecting PBS solution
D: Mixing the contents

**Quantity**
Question: How many pellets are gathered?
Options:
A: 10 pellets
B: 8 pellets
C: 12 pellets
D: 15 pellets

**Tool**
Question: Which tool is being used in this experimental step?
Options:
A: plastic bag
B: ziplock bag
C: desiccator
D: weigh boat

---

**Prompt for Level 2 Tasks**

**Sequence Generation**

**Task Instruction**

Solve the following question based on the video. Provide your final answer as a list of numbers (comma-separated) enclosed in \boxed{}.

**Question** Based on the full step list, determine the step numbers shown in the video.

Full Step List:

1. Use laryngoscope to expose vocal cords through mouth of 25–30g female Yorkshire pig
2. Spray vocal cords with two puffs of 2% lidocaine topical solution

. . .

39. Suture flap skin panel to cervical midline skin incision
40. Close abdominal skin incision

**Step Ordering**

**Task Instruction**

Solve the multiple choice question based on the video. Provide your final answer as a single letter enclosed in \boxed{}.

**Question** What is the correct sequence of steps shown in the video?

Options:

A: 1. Thoroughly mix equal proportions of epoxy and hardener
    2. Leave mixture for one hour

    . . .

B: 1. Place ZIF-8 membrane on 24mm steel disc with 5mm diameter center hole
    2. Thoroughly mix equal proportions of epoxy and hardener

    . . .

C: 1. Thoroughly mix equal proportions of epoxy and hardener
    2. Place ZIF-8 membrane on 24mm steel disc with 5mm diameter center hole

    . . .

D: 1. Thoroughly mix equal proportions of epoxy and hardener
    2. Leave mixture for one hour

    . . .

**Completeness Verification**

**Task Instruction**

Solve the multiple choice question based on the video. Provide your final answer as a single letter enclosed in \boxed{}.

**Question** Given the complete step list, which step was *not* performed in the video?

Step List:

1. Withdraw 1 milliliter of isoprene solution using syringe
2. Rinse syringe three times with isoprene solution prior to final withdrawal

. . .

7. Introduce flow of 2 standard liters per minute of purified air

Options:

A: 1    B: 2    C: 3    D: 4    E: 5    F: 6    G: 7

**Step Prediction**

**Task Instruction**

Solve the following question based on the video. Provide your final answer as a single number enclosed in \boxed{}.

**Question** Given all steps of the experiment, please predict the next operation that will take place after this video segment.

Full Step List:

1. Cut high purity copper foil into 4×4 cm pieces
2. Draw a line 0.5 cm from one edge of each square foil

. . .

44. Calculate permeance in Excel using mass spectrum data after steady state establishment

**Prompt for Level 3 Tasks**

**Full Prompt**
{task_instruction}
{question}

**Task Instruction**
Solve the following fill-in-the-blank question based on the video. Provide your final answer as a list of words or phrases (comma-separated) enclosed in \boxed{}.

**Question**

**Experimental Analysis**

Title: [(DPEPhos)(bcp)Cu]PF$_6$: A General and Broadly Applicable Copper-Based Photoredox Catalyst
Discipline: Chemistry
Question: In this study, the direct C2 arylation of N-methylpyrrole with 4-iodobenzonitrile at room temperature proceeded using ____ under ____ irradiation to afford the product in ____ yield after ____.

**Scientific Discovery**

Title: Novel Object Recognition Test for the Investigation of Learning and Memory in Mice
Discipline: Behavior
Question: In the mouse object recognition task, inhibitors of ____ produced a dose-dependent improvement in memory when administered 30 minutes before training, immediately after training, or 30 minutes before recall, implying that this target modulates ____, ____ and ____.

## F.4 ERROR CASES

We provide an output example of an Step Ordering Question from Gemini-2.5-Flash in both No-Thinking and Thinking_Budget=8192 modes. Since the full outputs are too lengthy, we present only key excerpts to highlight the distinct answering styles of the two modes.

---

**NoThinking Response (Correct, A):**

```
The video shows a precursor vaporization setup experiment.
The steps involved are:
1. Withdraw 1 milliliter of isoprene solution using syringe...
2. Rinse syringe three times with isoprene solution prior...
...
Let's analyze the options:
Option A: 1, 2, 3, 4, 5, 6, 7. This sequence aligns with the
logical progression of the experiment.
...
The video shows the following sequence of actions:
1. Preparation of the syringe (drawing and rinsing the solution).
2. Insertion of the syringe into the syringe pump.
3. Insertion of the syringe needle into the flask.
4. Heating of the flask.
5. Activation of the syringe pump.
6. Introduction of the carrier gas.

Comparing this with option A:
1. Withdraw 1 milliliter of isoprene solution using syringe. (Matches)
2. Rinse syringe three times with isoprene solution prior to final
withdrawal. (Matches)
...
The final answer is $\boxed{A}$....
```

**Thinking Response (Incorrect, D):**

```
In considering the question regarding the correct sequence of steps for
the Precursor Vaporization Setup experiment, I've reviewed the provided
options and the implicit logic of laboratory procedures for such setups.
My thought process has been to deconstruct each option, evaluate the
scientific rationale behind the order of operations, and ultimately
identify the most logical and efficient workflow.

Initially, I recognized that the experiment involves vaporizing a
precursor (isoprene solution) using a heated flask and a syringe pump
for controlled delivery, likely in conjunction with a carrier gas. This
immediately suggested a general workflow: prepare the sample, set up
the apparatus, prepare the reaction environment, introduce the sample,
and initiate the process.

I then systematically analyzed each proposed sequence:

**Option A**: I found this option plausible but noted potential issues.
Specifically, preheating the flask after inserting the syringe seemed
less efficient, and introducing the carrier gas as the very last step
raised concerns about optimal sweeping of the vaporized sample.
Considering the logical flow, D presents the most coherent sequence:
1.  Prepare the sample in the syringe.
2.  Place the syringe into the syringe injector.
3.  Preheat the flask.
4.  Insert the syringe needle into the flask.
5.  Introduce the carrier gas.
6.  Turn on the syringe injection.
...
The final answer is $\boxed{D}$....
```

# G  PROMPTS FOR ANNOTATION

In this section, we provide detailed prompts used in each annotation task.

## G.1  LEVEL-1

Prompts for Level-1 tasks are provided as follows.

### G.1.1  MATERIAL EXTRACTION

SYSTEM_PROMPT = "
You are an expert in scientific experimental procedure analysis, specializing in extracting **materials** from experimental procedure text. Please strictly follow the instructions by users.
"

USER_PROMPT_TEMPLATE = "
### Task Objective:
Extract the list of scientific **materials** mentioned in the following ASR transcript, preserving critical states and specifications.

You are given:
- An experimental step transcription (ASR caption): semantically accurate.
- A visual scene description from a vision-language model (Qwen caption): rough but helps verify visibility of the material.

### Material Definition:
- Biological specimens (with preparation state)
- Chemicals/reagents (with concentrations/forms)
- Solutions/mixtures (when specifically named)
- Gases/substrates

### Extraction Rules (Critical):
1. **Preserve essential descriptors** that define:
- Biological state (e.g., "anesthetized mouse", "fixed tissue")
- Preparation form (e.g., "trimmed hair", "lyophilized powder")
- Anatomical parts when manipulated (e.g., "mouse's head", "renal cortex")

2. Normalization guidelines:
- Keep singular/plural as in original context
- Remove non-essential modifiers (e.g., "carefully", "gently")
- Retain:
* Mixture states (e.g., "OVA-alum emulsified")
* Biological conditions (e.g., "post-mortem brain")

3. Exclusion criteria:
- Instruments/tools (e.g., "shaver", "pipette")
- Generic containers (e.g., "tube", "well plate")
- Unspecified solutions (e.g., just "solution")

### Output Format:
{ "materials": ["material1", "material2", ...] }

—

ASR caption: "{asr_caption}"
Qwen caption: "{qwen_caption}"
"

### G.1.2 MCQ Annotation for Material Recognition

SYSTEM_PROMPT = "
You are a scientific researcher creating multiple-choice questions (MCQs) for material recognition in scientific videos.
Your task is to generate 3 plausible distractors for a given material based on the experimental context.
"
USER_PROMPT_TEMPLATE = "
You are generating a multiple-choice question (MCQ) for material recognition in scientific experiment videos.
Given:
- An experimental step transcription (ASR): "{asr_caption}"
- A target material: "{target_material}"
—
### Your Task:
Generate **3 scientifically plausible distractors** (i.e., incorrect but believable options) for the given material.
### Each distractor must meet the following constraints:
1. Do not use distractors that only differ from the target material by quantity or concentration.

2. Must be an **actual material or chemical** used in real laboratory settings.

3. Must be **contextually plausible** in the described procedure — it should be reasonable that such a material might appear in this type of experiment.

4. Distractors should fall into **different plausible confusion categories**:
- **Visual similarity**: looks similar in appearance or form (e.g., transparent liquids)
- **Functional similarity**: used for similar purposes (e.g., washing, dissolving, blocking)
- **Common confusion**: frequently confused due to naming, function, or form
5. Do **not invent fake materials** or use vague terms (e.g., "solution", "fluid").
6. If the target material includes a modifier (e.g., "PBS buffer", "deionized water"), keep the full original phrase from the ASR as the correct answer.
—
Output ONLY valid JSON in the following format:
{
"question": "{question_template}",
"options": {
"A": "<correct answer with proper modifiers>",
"B": "<distractor 1>",
"C": "<distractor 2>",
"D": "<distractor 3>"
},
"answer": "A",
"target_material": "{target_material}",
"distractor_types": {
"B": "<visual/functional/confusion>",
"C": "<visual/functional/confusion>",
"D": "<visual/functional/confusion>"
}
}
Example for "PBS":
- A: "PBS" (correct)
- B: "saline solution" (functional - both for cell washing)
- C: "Tris buffer" (visual - similar buffer solutions)
- D: "deionized water" (confusion - commonly mistaken)
"

### G.1.3    TOOL EXTRACTION

---

SYSTEM_PROMPT = "
You are an expert in scientific experimental procedure analysis, specializing in extracting **tools** from experimental procedure text. Please strictly follow the instructions by users.
"

USER_PROMPT_TEMPLATE = "
### Task Objective:
Extract the list of scientific **tools** mentioned in the following ASR transcript.

You are given:
- An experimental step transcription (ASR caption): semantically accurate.
- A visual scene description from a vision-language model (Qwen caption): rough but helps verify visibility of the tool.

### Tool Definition:
Any instrument, equipment, or container used directly during the experiment (e.g., pipette, centrifuge, test tube).

### Standardization Rules:
1. Use lowercase and singular form (e.g., "gloves" → "glove").
2. Remove units or quantity descriptors (e.g., "1.5 milliliter microcentrifuge tube" → "microcentrifuge tube").
3. Remove generic adjectives or modifiers not affecting tool identity (e.g., "sterile", "clean"). Retain essential identifiers (e.g., "AVB Sepharose column").
4. Do not hallucinate. Only extract explicitly mentioned tools.

### Output Format:
{ "tools": ["tool1", "tool2", ...] }
—
ASR caption: "{asr_caption}"
Qwen caption: "{qwen_caption}"
"

---

### G.1.4 MCQ ANNOTATION FOR TOOL RECOGNITION

SYSTEM_PROMPT = "
You are a scientific researcher creating multiple-choice questions (MCQs) for tool recognition in scientific videos. Your task is to generate 3 plausible distractors for a given tool based on the experimental context.
"
USER_PROMPT_TEMPLATE = "
You are generating a multiple-choice question (MCQ) for tool recognition in scientific experiment videos.
Given:
- ASR: "{asr_caption}"
- Target tool: "{target_tool}"
—
Your task: Create 3 plausible distractors (wrong options) for the target tool.
### Requirements:
- Options must be tools that could reasonably appear in this experimental context.
- Distractors should be visually similar, functionally related, or commonly confused tools.
- If the target tool has modifiers (e.g., "microcentrifuge tube"), use the full phrase.
- Ensure the target tool name matches the ASR context.
### Output Format:
{
"question": "{question_template}",
"options": {
"A": "<correct answer with proper modifiers>",
"B": "<distractor 1>",
"C": "<distractor 2>",
"D": "<distractor 3>"
},
"answer": "A",
"target_tool": "{target_tool}",
"distractor_types": {
"B": "<visual/functional/confusion>",
"C": "<visual/functional/confusion>",
"D": "<visual/functional/confusion>"
}
}
Example for "pipette":
- A: "pipette" (correct)
- B: "syringe" (functional - both for liquid transfer)
- C: "dropper" (visual - similar appearance)
- D: "burette" (confusion - precise liquid measurement)
"

### G.1.5 QUANTITY RECOGNITION

SYSTEM_PROMPT = "
You are a scientific researcher creating multiple-choice questions (MCQs) for quantity recognition in scientific videos.
"

USER_PROMPT_TEMPLATE = "
You are generating a multiple-choice question (MCQ) for **quantity recognition** via visual observation.

You are given:
- An experimental step transcription (ASR caption).
- A visual scene description from a vision-language model (Qwen caption).
—

### Task:
Generate exactly ONE quantity-focused MCQ where the correct answer can only be determined by visually observing the video (e.g., volume, number of containers, temperature, duration).
—

### Rules:
1. Keep the question minimal and direct, focusing only on the quantity.
2. The answer must be visually inferable (use Qwen caption to check visibility).
3. Do not rely on textual or auditory clues.
—

### Distractor Guidelines:
- Options must be plausible in the context (realistic volumes, times, temperatures, counts).
- Keep distractors in the same magnitude range.
- Use visually confusable alternatives (e.g., 5 vs 7 tubes).
- Avoid overly fine distinctions (e.g., 5.0 vs 5.2 mL).
- Reflect common visual errors (slight miscounts, occlusion).
—

### Output Format:
{
"question": "<Clear, quantity-only question>",
"options": {
"A": "<correct answer>",
"B": "<distractor 1>",
"C": "<distractor 2>",
"D": "<distractor 3>"
},
"answer": "A"
}

If the ASR caption has no quantity-related info, return:
{ "question": null }

Input:
ASR caption: "{asr_caption}"
Qwen caption: "{qwen_caption}"
"

### G.1.6 OPERATION RECOGNITION (STAGE 1: ALIGNMENT SCORING)

USER_PROMPT_TEMPLATE = "
You will be given two inputs about the same video segment:
- ASR caption (narration of experimental steps)
- Qwen caption (vision-language description of the visual scene)
—

Your tasks:
1) Decide whether the segment contains experimental operation(s), preferably visible actions (e.g., pipetting, pouring, placing, transferring, cutting, mixing). If no experimental operation is present, or only background talking/intro without hands-on action, set the score to 0.
2) If operation(s) are present, judge the alignment between ASR and Qwen descriptions, and produce a score from 1 to 5 (higher = better alignment of actions/tools/entities/sequence).
—
Output JSON only with the fields:
{
"has_operation": <true—false>,
"visible_action": <true—false>,
"alignment_score": <integer 0-5>
}

Rules:
- If no operation: set has_operation=false, visible_action=false, alignment_score=0.
- If operations present: set has_operation=true; set visible_action=true only if the action is likely visible.
- For operations present: alignment_score in [1..5].

ASR caption: "{asr_caption}"
Qwen caption: "{qwen_caption}"
"

### G.1.7 OPERATION RECOGNITION (STAGE 2: MCQ GENERATION)

USER_PROMPT_TEMPLATE = "
You are generating a multiple-choice question (MCQ) for scientific experiment video understanding given the ASR caption.
—

### Task:
- Generate exactly ONE action-focused MCQ. The correct answer must describe a specific experimental operation stated in the ASR caption.
- Create 3 distractors that are plausible but incorrect variations of the action in the same tools/materials/setup context.
—
### Question design rules:
1. Minimal and direct: focus only on the observable action.
2. Visually grounded: the correct answer must be verifiable via video.
3. Do NOT use audio/textual clues (e.g., ASR narration). Assume only visual content is available.
—
### Distractor guidelines:
- Options must be plausible actions in the same context.
- Keep distractors in the same action/tool category (e.g., if pipetting is correct, distractors can be pouring, injecting, mixing).
- Avoid distractors that are too ambiguous or not visually distinguishable.
- Favor common mistakes or visually similar but incorrect operations (wrong hand, placing vs removing).
—
### Output Format:
{
"question": "<action-focused question strictly from ASR>",
"options": {
"A": "<correct action from ASR>",
"B": "<plausible distractor>",
"C": "<plausible distractor>",
"D": "<plausible distractor>"
},
"answer": "A"
}

ASR caption: "{asr_caption}"
"

## G.2 LEVEL-2

Prompts for Level-2 tasks are provided as follows.

### G.2.1 CLIP SEGMENTATION

SYSTEM_PROMPT = "
You are a scientific video annotation assistant. Your task is to segment a scientific experiment video transcript (ASR subtitles) into meaningful procedural clips for multi-step understanding benchmark.
"
USER_PROMPT_TEMPLATE = "
The benchmark focuses on medium-length videos containing several consecutive experimental steps. Each clip should:
- Include multiple related actions (usually 2+)
- Correspond to a coherent workflow unit (preparation, execution, wrap-up)
- Reflect logical/causal continuity
- Be suitable for designing multi-step reasoning questions
—
Please identify clip boundaries where:
- A major shift in experimental phase occurs
- The toolset, materials, or purpose changes significantly
- A natural grouping of steps can form a compact unit
—
### Output Format:
Return a JSON list where each segment has:
- "start_time": exact timestamp where the segment begins
- "end_time": exact timestamp where the segment ends
- "title": short summary of the clip
- "description": 1–2 sentences explaining the segment
—
### Rules:
1. Each segment must be 20–60 seconds long.
2. Start/end times must come directly from ASR (no invented timestamps).
3. Avoid over-segmentation of atomic actions; do not merge unrelated steps.
—
ASR transcript: "{asr_caption}"
"

### G.2.2 STEP EXTRACTION

SYSTEM_PROMPT = "
You are an expert in scientific experiment procedure analysis. Your task is to break down complex experimental procedures into atomic steps.
"

USER_PROMPT_TEMPLATE = "
You are an assistant tasked with decomposing scientific experiment procedures into atomic steps.
### Task:
Break down the experimental procedure in the timestamped subtitles into a sequence of **atomic steps**. Each step should represent a single action and include the corresponding time window.
### Guidelines:
- Only use the timestamped subtitles (ignore title/description).
- Each step must be: specific, self-contained, sequential, precise, and timed.
- Split compound actions into separate steps.
- Use technical language suitable for scientific protocols.
- If subtitles are ambiguous, make best effort with available info.
- If subtitles contain no experimental operation, return **null**.
### Output Format:
{
"atomic_steps": [
{
"step_number": 1,
"action": "<concise action description>",
"start_time": "<start_timestamp>",
"end_time": "<end_timestamp>"
}, ...
],
"total_steps": <integer>,
"confidence": "<high — medium — low>"
}
If no operations: return { null }.
### Example:
Timestamped Subtitles:
00:15.540 –> 00:19.140: Take 200 microliters of
00:19.140 –> 00:20.640: your culture of interest
00:22.590 –> 00:23.940: And just make a spot.
00:45.390 –> 00:46.080: I can usually
Expected Output:
{
"atomic_steps": [
{ "step_number": 1, "action": "Take 200 microliters of culture of interest", "start_time": "00:15.540", "end_time": "00:20.640" },
{ "step_number": 2, "action": "Make sample spots on plate", "start_time": "00:22.590", "end_time": "00:51.080" }
],
"total_steps": 3,
"confidence": "high"
}
Now analyze the given timestamped subtitles and generate atomic steps:
- Title: {title}
- Description: {description}
- Timestamped Subtitles: {timestamped_subtitles}
"

### G.2.3 SEQUENCE ORDERING MCQ

SYSTEM_PROMPT = "
You are an expert in creating multiple-choice questions for scientific experiment step sequencing.
"
USER_PROMPT_TEMPLATE = "
You are creating a multiple-choice question about the correct sequence of experimental steps.
—
### Context:
- Title: {title}
- Description: {description}
—
### Task:
Given the following correct sequence of atomic steps, create an MCQ with 4 options (A, B, C, D):
- Option A = correct sequence
- Options B, C, D = incorrect but plausible sequences
—
### Correct Sequence:
{steps_text}
—
### Requirements for incorrect options:
1. Maintain scientific plausibility.
2. Keep logical procedural flow (no impossible orders).
3. Introduce subtle ordering variations (swap/rearrange steps plausibly).
4. Use the same steps — only reorder.
—
### Output Format:
{
"question": "What is the correct sequence of steps for this experimental procedure?",
"options": {
"A": "1. <correct step 1>2. <correct step 2>3. <correct step 3>...",
"B": "1. <incorrect step 1>2. <...>",
"C": "1. <incorrect step 1>2. <...>",
"D": "1. <incorrect step 1>2. <...>"
},
"correct_answer": "A"
}

Generate the MCQ now.
"

