# OpenReview forum: "ExpVid: A Benchmark for Experiment Video Understanding & Reasoning"
_ICLR.cc/2026/Conference — ICLR 2026 Poster_

### Official Review · Reviewer_QvUU · 2025-10-26

**Soundness:** 3
**Presentation:** 2
**Contribution:** 3
**Rating:** 6
**Confidence:** 3

**Summary:**

This paper evaluates the extent to which multimodal large language models (MLLMs) (which have the capacity to analyze videos) can be used for important tasks in wetlab settings. This includes identifying different aspects of experimental tasks from video, understanding sequences of tasks that make up a particular procedure, and drawing conclusions from a sequence of scientific procedures. To this end, the paper introduces a new dataset ExpVid, which aims to capture some of the capabilities required for a video-processing MLLM to be useful in a wetlab setting. The paper breaks these capabilities up into a hierarchy: finegrained perception, procedural understanding, and scientific reasoning. ExpVid contains questions aimed at each of these. After describing how the dataset was constructed, automatically annotated, and verified, it provides the results of evaluating both closed and open models. Broadly, it is found that closed models like GPT5 and Gemini have substantially better performance. The paper ends with some further light analysis of the results.

**Strengths:**

**Clarity:** In general, the paper is well-written (but see typos in the Nitpicks section below and the issue of missing examples). It is clear that a lot of work was put into the figures, which are clean and communicate well the points of the paper. The reviewer believes that most participants at ICLR will be able to read and appreciate the results while on the other hand, a good amount of detail on the dataset generation process (usual to experts) was included.

**Problem importance:** The use of MLLMs (not restricted to the video models in this paper) in science is a growing area with several notable start-ups focused on this appearing in the US just this year. Getting a better understanding of the real capabilities of these models in this space is thus highly relevant to current developments in the field. In this sense, benchmarks such as ExpVid, are a welcome contribution.

**Weaknesses:**

**Without access to the videos, it is hard for a reader to gauge the difficulty of these tasks:** Though the reproducibility statement says that an anonymized copy of the dataset would be made available, the reviewer did not see this (please correct me if I just missed it). Without this, it is hard to get a good sense of the actual content of the dataset and therefore challenging to know how to interpret the results. The reviewer would also recommend adding more examples of questions from the dataset (as much as this is possible for a text and image-based paper) in the body of the paper itself. One way to do this would be to add example questions corresponding to each of the categories introduced in Section 3.2.

**The paper would have benefited from deeper analysis of the results:** It is challenging to include all the details in a dataset paper that has a page limit of only 9 pages. Especially when dataset construction was complicated with many different steps (as with this paper). However, thinking about what is most valuable to the average reader, this reviewer believes there is a strong argument for including more analysis of the results, possibly at the expense of moving some details of dataset construction to the appendix. In particular, the reviewer was curious about current limitations of all models in the tasks that the paper captures. What are limitations that the model-building part of the community needs to address? Where are we succeeding?

**Nitpicks**
- Line 121: “…provide their corresponding analysis…” $\mapsto$ “…provide an analysis of our results…”?
- Line 158: “…while most of it in computation and physics are…” $\mapsto$ “…while most computational examples, or examples from physics…”?
- Line 195: “Whether records an entire…” missing ‘it’.
- Line 261: The term ‘MCQ’ is used before it is defined.
- Line 264: “Identify the appeared tools from the scene…” should be “tools that appear” or something like that.
- Line 323: “Encouraging multi-blank settings to probing several key points within one question.” What does this mean?

**Questions:**

- The paper says that an anonymized version of the dataset would be released, but the reviewer could not find any links to this?
- One of the tasks is ‘quantity recognition’. This reviewer has trouble identifying numbers appearing on devices in a video. How was it verified that this task was feasible? Examples of what this looks like would be very helpful.
- Line 463: How was the ablation of frames performed?

---

> ### Author Response · Authors · 2025-11-21
>
> We thank the reviewer for the positive feedback and address the concerns as follows.
>
> ---
> >***W1&Q1** Without access to the videos, it is hard for a reader to gauge the difficulty of these tasks*
>
> **A1** Thank you for the suggestion. We have added several figures of example items for all task types in **Appendix D (p.17, line 864)**. We also provide a subset of videos with corresponding annotations in an anonymized repository, referenced at **Appendix D, line 868**. Due to the page limitation, we will include representative QA examples in Section 3.2 in the camera-ready version according to your suggestion.
>
>
> ---
> >***W2** The paper would have benefited from deeper analysis of the results; What are limitations that the model-building part of the community needs to address? Where are we succeeding?*
>
> **A2** Thank you for the suggestion. We have added the following analysis.
>
> We average performance across all evaluated models on each task as follows.
>
> |Tool|Mat.|Quan.|Oper.|L1 Avg.|Ord.|Gen.|Veri.|Pred.|L2 Avg.|Anal.|Disc.|L3 Avg.|
> |----|----|-----|-----|------|----|----|-----|-----|------|-----|-----|------|
> |35.9|34.3|53.6|67.0|**47.2**|68.8|35.3|22.4|17.8|**36.1**|33.2|30.1|**31.6**|
>
> Comparing the three levels, models perform substantially better on Level-1 short-clip visual recognition than on longer-horizon procedural understanding in Level-2 and 3. This suggests that current MLLMs are stronger at **general image understanding**, especially in Quantity and Operation, where performance benefits from capabilities such as OCR and human activity recognition accumulated through large-scale vision–language pretraining. In contrast, there is clear room for improvement on **science-specific visual elements**, particularly the recognition of materials and tools in laboratory environments.
>
> The much lower scores on Level-2 and Level-3 further indicate that understanding complex procedural experiment videos remains a major challenge. Within Level-2, the relatively high performance on Ordering suggests that models can leverage strong prior procedural knowledge (from large-scale text pretraining on scientific content) when the task is less dependent on visual evidence. However, for the other Level-2 tasks, performance drops markedly once the answer must be based on evidences grounded in the given video. This highlights limitations in **(i) the ability of understanding and reasoning over long-range multi-step videos** and (ii) **the scarcity of training data for lab experiment videos**.
>
> Following your advice, we will integrate this discussion and the corresponding table into the main body of the paper in the camera-ready version.
>
> ---
> >***W3** Nitpicks*
>
> **A3** Thanks. We have corrected the phrasing (L121, L158, L195, L264), defined MCQ at first mention (L261), and clarified the multi-blank guideline (L323) as "Should annotate multi-blank questions, where each question contains multiple blanks that capture several key information points". All corrections are highlighted in the uploaded paper.
>
> ---
> >***Q2** Verifying the feasibility of 'Quantity recognition'.*
>
> **A4** In expert verification, an explicit criterion is that annotators must ensure the target quantity is either **visibly observable** in the video or can be **inferred from the experimental visual context**. Items that do not meet this requirement are removed. We provide several examples of Quantity Recognition questions in the newly added **Figure 7 in Appendix D (p.17)**, including the relevant visual evidence and the annotators’ justification reasons. These examples demonstrate that the task is indeed feasible.
>
> ---
> >***Q3** How was the ablation of frames performed?*
>
> **A5** The model input consists of uniformly sampled frames from the full video. The ablation is performed by varying **the number of sampled frames** used as input. We will clarify this procedure in the ablation study section of the paper.
>
> ---
> We hope the clarifications above address the concerns and contribute positively to your evaluation. Please let us know if there are further refinements that would help you consider raising the score.

---

### Official Review · Reviewer_stVf · 2025-10-31

**Soundness:** 2
**Presentation:** 2
**Contribution:** 2
**Rating:** 6
**Confidence:** 3

**Summary:**

The paper introduces ExpVid, a new benchmark designed to evaluate Multimodal Large Language Models (MLLMs) on scientific experiment videos, focusing on wet-lab procedures across 13 disciplines. ExpVid is structured around a three-level task hierarchy: (1) Fine-grained Perception, (2) Procedural Understanding, and (3) Scientific Reasoning. The benchmark includes 390 curated experiment videos with detailed annotations from peer-reviewed publications. The study evaluates 19 MLLMs, revealing their strengths in coarse-grained recognition but highlighting limitations in fine-grained tasks and scientific reasoning.

**Strengths:**

* **Multilevel Evaluation Capacity:** ExpVid introduces a three-tier task hierarchy: fine-grained perception (identifying tools, materials, and actions), procedural understanding (step order and completeness), and scientific reasoning (linking experiments to conclusions). This ensures comprehensive assessment of MLLMs' capabilities across varying complexity levels.

* **Scientific and Realistic Data:** The videos in ExpVid are sourced from the peer-reviewed JoVE journal, ensuring scientific accuracy and reflecting actual experimental processes. The task design closely aligns with real-world experimental workflows, avoiding overly theoretical tests.

* **Vision-Centric Annotation with Expert Validation:** ExpVid uses a vision-based annotation pipeline combined with expert review, minimizing reliance on text knowledge and enhancing the reliability and diversity of tasks. This process ensures accurate and contextually appropriate task creation.

**Weaknesses:**

1. **Input Format Ambiguity:** The paper does not clearly specify the input format of the experiment videos used in the evaluation. It is important to clarify whether the inputs are full videos, images extracted from videos at intervals, or something else. This detail could significantly impact the model's performance, as different input formats may require different processing approaches. Clearer descriptions of the input format would enhance reproducibility and understanding of the benchmark.

2. **Expert Annotation Scale and Feasibility:** While the paper highlights the use of PhD-level expert annotators, the number of experts and their workload distribution across various disciplines are not provided. Given the breadth of scientific fields covered by ExpVid, it would be useful to understand how many experts were involved, how much time they spent, and how the workload was managed across such a diverse set of tasks. This information would help assess the scalability and practicality of the annotation process, which could be a potential bottleneck for future applications of this benchmark.

3. **Typographical Errors:** A minor but notable typographical error occurs on line 282, where "e.g." is incorrectly formatted as "e.g." without a following comma. Ensuring that such errors are corrected would improve the overall presentation of the paper.

**Questions:**

1. **Clarification on Input Format:**

   * Could you clarify the exact input format used in ExpVid? Specifically, are the models provided with full videos, or are the videos split into individual frames or intervals of frames? How does this format affect model performance, and have you considered experimenting with different formats to evaluate potential performance variations?

2. **Expert Annotation Workload:**

   * Given the wide range of scientific disciplines covered in ExpVid, could you provide more details on the number of PhD-level expert annotators involved in the project? How were the tasks distributed across experts, and how much time did they spend on annotation? This information would help understand the scalability of the annotation process and any potential limitations.

---

> ### Author Response · Authors · 2025-11-21
>
> We thank the reviewer for the positive feedback and address the concerns as follows.
>
> ---
> >***W1&Q1** Input Format Ambiguity & Clarification on Input Format*
>
> **A1** Thanks for raising this issue. We clarify that all models are evaluated using **uniformly sampled frames from the full video**, and only these sampled frames are fed into the model. The number of frames differs by task level: Level-1 uses 8 frames, Level-2 uses 64 frames (both roughly 1 fps). Level-3 uses model-specific frame counts, as detailed in **Table 3 in Appendix F.3 (p.25, line 1301)**. We will add this clarification to the experimental setup section in the main paper in the camera-ready version.
>
> ---
> >***W2&Q2** Expert Annotation Scale and Feasibility & Expert Annotation Workload*
>
> **A2** To clarify the expert annotation workload and how it was managed across diverse scientific domains, we summarize the process below.
>
> 1) **Personnel Capacity.** We maintained a pool of 15 domain-prepared annotators per major category (e.g., medicine, biology) and a total team size of 50 annotators, enabling timely task distribution across disciplines.
>
> 2) **Annotation Interface.** To support consistent and scalable annotation across these heterogeneous tasks, we built a dedicated annotation platform with task-specific interfaces tailored to the requirements of each question type. These interfaces guide annotators through the rubric and enforce correct application of the annotation schema. Importantly, every annotation requires the annotator to provide a justification, even when simply approving a question, ensuring transparency and quality control. Examples of the interfaces of different tasks are provided in Appendix Section D.
>
> 3) **Criteria.**
> - **Video-grounded**: All questions must be solvable from the visual evidence in the video.
> - **No leakage or guessing shortcuts**: Stems must not reveal answers; distractors must be scientifically plausible.
> - **Concrete, step-level fidelity**: Only visually verifiable actions are retained; abstract or unobservable descriptions are removed or rewritten.
> - **Consistent formatting and clarity**: Wording avoids unverifiable details and ensures each question has a unique, unambiguous answer.
> - **Justified verification**: Annotators provide a brief rationale for every accepted or corrected item to ensure transparency.
>
> 4) **Time Cost**. Each annotator first reviews the full experiment video together with its accompanying paper, which takes approximately 40 minutes per item. The subsequent annotation times vary by task type:
> - **Level-1**: ~6–8 minutes per question
> - **Level-2**: ~13 minutes per question
> - **Level-3**: ~18 minutes per question
>
> The full expert annotation process consisted of a one-month pilot phase, used for iterative feedback and guideline alignment, and then followed by one month of formal annotation to complete the benchmark.
>
> We have added the aforementioned Expert Verification Workflow into **Appendix E, page 20** of the revised manuscript.
>
> ---
> >***W3** Typographical Errors*
>
> **A3** Thank you for pointing this out. We will fix the ‘e.g.,’ and run a full style pass to eliminate typos.
>
> ---
> Based on these clarifications, we hope you could consider increasing your score in support of our work. Otherwise, could you let us know any additional changes you would like to see in order for this work to be accepted?

---

### Official Review · Reviewer_1vC1 · 2025-10-31

**Soundness:** 4
**Presentation:** 3
**Contribution:** 3
**Rating:** 8
**Confidence:** 4

**Summary:**

This paper introduces ExpVid, a new and significant benchmark for evaluating Multimodal MLLMs on scientific experiment video understanding. Its primary contribution lies in addressing a clear gap, as existing benchmarks fail to capture the fine-grained, long-horizon nature of "wet-lab" procedures. Its core contribution is a three-level task hierarchy (Perception, Procedure, Reasoning) which allows for a detailed diagnosis of model capabilities, from basic visual recognition up to complex, long-horizon inference and is built from peer-reviewed JoVE videos paired with their scientific publications. This unique data pairing enables high-level reasoning tasks absent in other datasets. Crucially, the authors employ a rigorous "vision-centric" annotation pipeline with PhD-level expert validation, ensuring that tasks require genuine visual grounding and cannot be "cheated" by models relying on prior data. Finally, the paper provides actionable findings by evaluating 19 SOTA MLLMs, clearly demonstrating their current limitations and offering a valuable roadmap for future research.

**Strengths:**

- The three-level task hierarchy is a major contribution. This structure is exceptionally well-conceived, as it mirrors the real-world scientific process—from basic observation of tools to high-level reasoning about an experiment's conclusions. This hierarchy allows for a much more granular diagnosis of model capabilities than a single, monolithic task, enabling researchers to pinpoint where models are failing.
- The choice of data source (peer-reviewed JoVE videos) is excellent. By pairing videos with their corresponding peer-reviewed publications, the authors ensure a high degree of scientific rigor.
- The "vision-centric" annotation pipeline is a critical methodological strength. The authors explicitly designed the tasks to require visual grounding, using semantically and visually plausible distractors. By validating these tasks with multi-disciplinary, PhD-level experts, they have addressed the well-known "language-prior shortcut" problem (where models guess the answer from text without processing the video).
- The paper provides more than just a dataset; it delivers a strong empirical study by benchmarking 19 leading MLLMs. The findings are clear and valuable: all models struggle with fine details and long-horizon reasoning, and a significant performance gap exists between proprietary and open-source models.

**Weaknesses:**

- The benchmark is built on 390 high-quality videos. While the annotation depth (7,800 QA pairs) is impressive, the total number of unique experimental videos is relatively small compared to other large-scale video benchmarks.
- The data is sourced from JoVE. This is a strength for rigor but a potential weakness for real-world applicability. JoVE videos are highly produced, professionally narrated, and edited for clarity. They do not represent the "messy" reality of lab work, which often involves poor lighting, occlusions, camera-phone footage, verbal mistakes, and actual failed steps. Models trained for this benchmark may not generalize well to the real lab videos.
- The Level-3 (Scientific Reasoning) tasks are evaluated using a "fill-in-the-blank" format. This Outcome-Based method effectively tests the outcome of a model's reasoning but fails to evaluate the process. As the authors acknowledge, it doesn't illuminate the chain-of-thought. A model could arrive at the correct answer through flawed logic, or fail for a simple perceptual mistake. This is a missed opportunity for a benchmark on "reasoning."
- The paper uses non-expert undergraduates as a human baseline. This baseline is difficult to interpret. One would expect non-experts to fail at Level-3 (Scientific Reasoning), making their (unreported) score on this task less informative. A more valuable, albeit much more difficult, baseline would have been to benchmark the performance of PhD-level experts to understand the true gap between models and expert-level scientific comprehension.

**Questions:**

- The Level-3 tasks are "outcome-based" (fill-in-the-blank), which checks if the final answer is correct but not if the reasoning process was. How can we be sure models are not arriving at correct answers through flawed logic? Was a "process-based" (e.g., chain-of-thought) evaluation considered, and why was it excluded?
- The paper makes a point that "Thinking" mode can hurt performance by causing models to "drift" from visual evidence. This is a critical but under explained finding. Could the authors elaborate on how prevalent this was? Does this suggest a fundamental flaw in how current models balance their internal knowledge against real-time visual evidence?
- The paper states ASR transcripts were collected from JoVE. Is there any data on the quality (e.g., Word Error Rate) of these transcripts? It is ambiguous whether some model failures, especially on Level 2 and 3, could be attributed to errors in the provided ASR text rather than the model's own reasoning.

---

> ### Author Response · Authors · 2025-11-21
>
> We thank the reviewer for the positive feedback and address the concerns as follows:
>
> ---
> >***W1** Relatively small scale of unique experimental videos*
>
> **A1** The initial release contains 390 videos primarily due to the high cost of expert verification. We view this as a first version and will continue to expand the benchmark with additional disciplines and more experiments in future iterations.
>
> ---
> >***W2** Potential weakness for real-world applicability*
>
> **A2** We agree with the reviewer’s point. The highly produced and professionally narrated nature of JoVE videos is precisely what enables us to obtain accurate step-level explanations and paired articles for high-quality annotation. In contrast, constructing reliable annotations from the “messier” and more diverse real-world lab footage is substantially harder and requires high cost.
>
> Importantly, despite being sourced from highly produced JoVE videos, our benchmark remains very challenging for current state-of-the-art MLLMs shown in our evaluation results, especially on tasks that require understanding and reasoning over multi-step experimental procedures. This demonstrates that our benchmark serves as a strong foundation and stepping stone toward AI systems for scientific experiments. We view it as a necessary first stage before generalizing to more challenging, messy reality of lab work.
>
> ---
> >***W3&Q1** Fail to evaluate the reasoning process in Level-3 tasks*
>
> **A3** We agree that evaluating the full reasoning process is valuable, but annotating ground-truth chains-of-thought for scientific experiments is prohibitively costly and often ambiguous, even for experts. Given the already substantial annotation burden, we adopt an outcome-based fill-in-the-blank format because it provides **objective, unbiased scoring** without relying on free-form answer matching.
>
> Importantly, this format still requires procedural reasoning over the full video: the blanks cannot be answered from the text alone or from a single frame. Annotators design stems that avoid leaking the answer, ensuring that the model must integrate visual evidence across multiple steps of the experiment. As shown in Figure 3 “Effect of Input Video Frames” right (Page 9), when only the text is provided without any video input, model performance degrades to below 10%. This confirms that our design indeed requires reasoning grounded in the complete experiment rather than text-only shortcuts. Thus, while we do not annotate explicit CoT, the current design still enforces genuine reasoning over the observed procedure.
>
> As noted in the paper’s Limitation section, Level-3 currently evaluates outcomes rather than the underlying reasoning process. Extending the benchmark toward more explicit reasoning evaluation is a key direction for future work.
>
> ---
> >***W4** Non-expert undergraduates as the human baseline*
>
> **A4** We agree. While a PhD-level baseline must come from experts not involved in annotation, we could not reuse our annotators. We plan to recruit an independent group of PhD-level participants to establish a proper expert baseline in future updates.

---

> ### Author Response · Authors · 2025-11-21
>
> >***Q2** Issue of thinking effect*
>
> **A5** We have added comparison experiments between thinking and non-thinking modes, as well as different thinking-budget settings across three model families, as shown below.
>
> |Model|Thinking|Tool|Mat.|Quan.|Oper.|L1Avg.|Ord.|Gen.|Veri.|Pred.|L2Avg.|Anal.|Disc.|L3Avg.|
> |-----|-----|----|----|-----|-----|------|----|----|-----|-----|------|-----|-----|------|
> |MiMo-VL-7B-RL|No|34.2|33.7|44.2|62.4|42.4|43.9|28.5|18.5|11.4|27.4|28.7|25.9|27.3|
> |MiMo-VL-7B-RL|Yes|36.1|29.1|53.6|67.8|44.3|64.8|32.3|24.9|15.6|34.3|29.3|27.3|28.3|
> |Gemini-2.5-Flash|No|52.7|50.1|65.2|72.6|58.6|86.0|50.5|24.1|40.2|50.1|47.2|41.1|44.1|
> |Gemini-2.5-Flash|Yes|52.7|50.7|71.9|73.3|60.2|85.1|54.3|22.3|38.0|49.8|44.8|41.3|43.0|
> |Gemini-2.5-Pro|No|53.1|45.9|64.3|80.8|59.2|83.7|61.3|26.8|49.6|53.8|50.6|45.2|47.9|
> |Gemini-2.5-Pro|Yes|51.3|44.3|63.8|74.4|56.7|84.2|59.9|26.8|46.9|54.3|50.1|44.8|47.4|
> |Qwen3-VL-235B-A22B-Instruct|No|30.2|34.1|50.0|68.5|45.1|74.0|42.3|16.2|19.6|39.0|40.5|31.8|36.1|
> |Qwen3-VL-235B-A22B-Thinking|Low|35.7|30.2|48.6|68.8|43.9|69.9|36.6|20.3|24.3|37.8|30.2|36.3|33.1|
> |Qwen3-VL-235B-A22B-Thinking|Mid|35.7|33.3|45.7|74.2|45.6|68.5|39.1|14.9|21.6|36.0|39.6|37.4|38.5|
> |Qwen3-VL-235B-A22B-Thinking|High|36.6|31.0|48.6|71.0|44.9|68.5|38.3|21.6|27.0|38.9|32.2|38.1|35.1|
>
> From the results, we observe that longer thinking does not yield the similar improvements seen in language-only tasks, where test-time scaling often benefits long-context reasoning. On video tasks, the thinking effect remains limited, which supports our claim that thinking does not consistently improve results. The patterns from Qwen3-VL under different thinking budgets further show that a high thinking budget can even reduce accuracy on several tasks.
>
> As illustrated in the error cases in Appendix E.5, excessive thinking can cause models to over-reason, rely too heavily on prior knowledge, and search for the “most logical” workflow rather than grounding their answers in the actual visual evidence. This drift from the video often leads to hallucinated results and is a key source of errors in video-grounded QA.
>
> As also noted in Qwen3-VL’s technical blog [1], the performance of the Thinking mode on video benchmarks is mixed relative to the Instruct mode and does not show a consistent improvement. Across different benchmarks and different models, this indicates a prevalent pattern that Thinking provides limited benefit on video-grounded tasks.
>
> We will add the additional results and discussion to Section 4.2 More Analysis in the camera-ready version.
>
> ---
> >***Q3** Potential low-quality ASR transcripts lead to model failures; data on the quality of transcripts*
>
> **A6** Low-quality ASR transcripts do not contribute to model failures in our benchmark. Although some JoVE transcripts do contain errors, we filter them out during data curation. As described in **Sec. 3.1 (line 191)**, we run an LLM-based filtering by five criteria, and only select videos from whose ASR meets all requirements. The transcript-quality statistics of 15k candidates are shown below.
>
> |Criterion|Mean|Std|
> |-|----|----|
> |Continuity|4.287|0.563|
> |Clarity|4.807|0.438|
> |Focus|4.459|0.570|
> |Integrity|3.506|0.709|
> |Alignment|3.490|0.962|
> |Overall Score|4.110|0.508|
>
> Furthermore, if any residual ASR issues inadvertently influence the generated questions, they are corrected or removed during human verification, ensuring that the final benchmark does not include annotations whose difficulty stems from ASR noise.
>
> ---
> Based on these clarifications, we hope you can keep your support of this work.
>
> ---
> ***Reference***
>
> [1] Qwen3-VL: Sharper Vision, Deeper Thought, Broader Action, https://qwen.ai/blog?id=99f0335c4ad9ff6153e517418d48535ab6d8afef&from=research.latest-advancements-list

---

### Official Review · Reviewer_Z8zB · 2025-10-31

**Soundness:** 2
**Presentation:** 3
**Contribution:** 2
**Rating:** 6
**Confidence:** 4

**Summary:**

ExpVid introduces a benchmark for understanding and reasoning over scientific experiment videos, with a three-level hierarchy: (L1) fine-grained perception (materials, tools, quantities, operations), (L2) procedural understanding (ordering, sequence generation, completeness, next-step prediction), and (L3) scientific reasoning (experimental analysis, discovery). The dataset is curated from peer-reviewed research collection videos and paired papers, using a vision-centric, semi-automatic pipeline (LLM-assisted extraction + expert verification). The released scale is 390 videos across 13 disciplines with 7,800 QA items spanning 10 task types. Baselines across 19 MLLMs show frontier closed-source models outperform open-source ones, with the gap widening at higher-order reasoning; e.g., GPT-5 leads L2/L3, and Gemini-2.5-Flash tops L1. Authors report vision is necessary (ablations w/ vs. w/o frames) and “thinking” can sometimes hurt if it drifts from visuals. Limitations include a focus on wet-lab domains and L3 evaluation via an LLM judge rather than human-graded answers.

**Strengths:**

1. The task design is clear and hierarchical (perception $\rightarrow$ procedure $\rightarrow$ reasoning), aligning with practical laboratory steps and utilizing expert-validated task templates and distractors.
2. The paper details a transparent curation process, utilizing JoVE videos and expert verification, to create a substantial and diverse benchmark (7,800 QA pairs across 13 disciplines).
3. The paper provides a comprehensive empirical evaluation by testing 19 MLLMs against human baselines using task-appropriate metrics for comparison.

**Weaknesses:**

1. The Level-3 (Scientific Reasoning) tasks use a "fill-in-the-blank" format (Sec 3.3). This format might be simpler or more susceptible to text-matching than open-ended questions. Please discuss the rationale for this choice and how it was designed to test deep reasoning beyond simple keyword extraction.
2. The Thinking effect observation is insightful but based on one model configuration and qualitative examples. Please Run a 2×2 across ≥3 model families with or w/o think within matched budgets to verify the claim.
3. L2 shows high Step Ordering yet low Step Prediction/Completeness, but diagnostics aren’t tied to controllable factors (occlusion, clip length, distractor difficulty). Please provide controlled L2 suites that vary occlusion, temporal gap, and distractor closeness, and report item-response curves.

**Questions:**

N/A

---

> ### Author Response · Authors · 2025-11-21
>
> We appreciate the reviewer's positive feedback on our work. Below, we address the reviewer's concerns in turn:
>
> ---
> >***W1** The rationale for choosing “fill-in-the-blank” format for Level-3 tasks.*
>
> **A1** Our choice of fill-in-the-blank format is motivated by the need to evaluate a model’s ability to bridge what it observes in the experiment video to the outcome analysis and scientific conclusion, while keeping the evaluation reliable and unbiased.
>
> 1) **Reliable scoring without linguistic variance.**
>  Pure open-ended QA produces highly diverse answers, making it difficult to determine whether the model correctly identifies the key conclusion derived from the experimental procedure. Using LLM-as-a-judge for these free-form answers would introduce additional bias. By contrast, the fill-in-the-blank format provides controlled generation, eliminating linguistic variability and allowing us to directly assess whether the model arrives at the correct conclusion grounded in the video.
>
> 2) **Depends on reasoning over full-procedure visual evidence.**
> Even with a constrained output format, the blank cannot be answered from the text alone, prior domain knowledge, or a keyword in a single frame. Annotators design stems that do not leak the answer, forcing the model to rely on observations across the full experimental workflow, for example, comparing before/after conditions, interpreting observed outcomes, or synthesizing multi-step procedural cues. We also avoid questions solvable by looking at a single frame; instead, the required information spans multiple steps or the complete procedure.
> As shown in Figure 3 “Effect of Input Video Frames” right (Page 9), when only the text is provided without any video input, model performance degrades to below 10%. This confirms that our design indeed requires reasoning grounded in the complete experiment rather than text-only shortcuts.
>
> ---
> >***W2** Thinking effect observation*
>
> **A2** Thank you for the suggestion. We have added comparison experiments between thinking and non-thinking modes, as well as different thinking-budget settings across three model families, as shown below.
>
> |Model|Thinking|Tool|Mat.|Quan.|Oper.|L1Avg.|Ord.|Gen.|Veri.|Pred.|L2Avg.|Anal.|Disc.|L3Avg.|
> |-----|-----|----|----|-----|-----|------|----|----|-----|-----|------|-----|-----|------|
> |MiMo-VL-7B-RL|No|34.2|33.7|44.2|62.4|42.4|43.9|28.5|18.5|11.4|27.4|28.7|25.9|27.3|
> |MiMo-VL-7B-RL|Yes|36.1|29.1|53.6|67.8|44.3|64.8|32.3|24.9|15.6|34.3|29.3|27.3|28.3|
> |Gemini-2.5-Flash|No|52.7|50.1|65.2|72.6|58.6|86.0|50.5|24.1|40.2|50.1|47.2|41.1|44.1|
> |Gemini-2.5-Flash|Yes|52.7|50.7|71.9|73.3|60.2|85.1|54.3|22.3|38.0|49.8|44.8|41.3|43.0|
> |Gemini-2.5-Pro|No|53.1|45.9|64.3|80.8|59.2|83.7|61.3|26.8|49.6|53.8|50.6|45.2|47.9|
> |Gemini-2.5-Pro|Yes|51.3|44.3|63.8|74.4|56.7|84.2|59.9|26.8|46.9|54.3|50.1|44.8|47.4|
> |Qwen3-VL-235B-A22B-Instruct|No|30.2|34.1|50.0|68.5|45.1|74.0|42.3|16.2|19.6|39.0|40.5|31.8|36.1|
> |Qwen3-VL-235B-A22B-Thinking|Low|35.7|30.2|48.6|68.8|43.9|69.9|36.6|20.3|24.3|37.8|30.2|36.3|33.1|
> |Qwen3-VL-235B-A22B-Thinking|Mid|35.7|33.3|45.7|74.2|45.6|68.5|39.1|14.9|21.6|36.0|39.6|37.4|38.5|
> |Qwen3-VL-235B-A22B-Thinking|High|36.6|31.0|48.6|71.0|44.9|68.5|38.3|21.6|27.0|38.9|32.2|38.1|35.1|
>
> From the results, we observe that longer thinking does not yield the similar improvements seen in language-only tasks, where test-time scaling often benefits long-context reasoning. On video tasks, the thinking effect remains limited, which supports our claim that thinking does not consistently improve results. The patterns from Qwen3-VL under different thinking budgets further show that a high thinking budget can even reduce accuracy on several tasks. As also noted in Qwen3-VL’s technical blog [1], the performance of thinking mode on video benchmarks is mixed compared to the Instruct mode and does not show a consistent improvement.
>
> As illustrated in the error cases in Appendix E.5, excessive thinking can cause models to over-reason, rely too heavily on prior knowledge, and search for the “most logical” workflow rather than grounding their answers in the actual visual evidence. This drift from the video often leads to hallucinated results and is a key source of errors in video-grounded QA.
>
> We will add the additional results and discussion to Section 4.2 More Analysis in the camera-ready version.
>
> ---
> ***Reference***
>
> [1] Qwen3-VL: Sharper Vision, Deeper Thought, Broader Action, https://qwen.ai/blog?id=99f0335c4ad9ff6153e517418d48535ab6d8afef&from=research.latest-advancements-list

---

> > ### Author Response · Authors · 2025-11-21
> >
> > >***W3** L2 high Step Ordering yet low Step Prediction/Completeness. Diagnostics not factorized*
> >
> > **A3** We analyze the performance gap between Step Ordering and Step Prediction/Completeness as follows.
> >
> > 1. **Number of distractors**. Step Ordering has only four candidate steps, whereas Step Prediction and Completeness involve larger candidate sets. We report the average number of candidate steps for each task:
> >   - Sequence Ordering: option_count = 4.0 (std = 0.00)
> >   - Video Verification: option_count = 6.2 (std = 2.50)
> >   - Step Prediction: option_count = 48.4 (std = 11.97)
> >
> > Correspondingly, we present the item–response curve using performance averaged over all evaluated models. It clearly shows that tasks with higher distractor counts exhibit lower model performance.
> >
> > |Option count|Model performance|
> > |------------|-----------------|
> > |4.0|68.8|
> > |6.2|23.4|
> > |48.4|18.0|
> >
> > 2. **Ordering benefits from prior procedural knowledge**. Ordering is not a counterfactual task, where many sequences follow common-sense experimental logic. Even though we design distractors as plausible as possible, models can often select the most reasonable step based on prior knowledge alone. In contrast, for Prediction and Completeness, the target-step segment is removed from the input. Although this does not significantly change the video’s temporal scale, it forces the model to rely on visual grounding in the remaining evidence rather than general procedural priors, making these tasks inherently more challenging.
> >
> > We will include this analysis in Section 4 Experiment in the camera-ready version.
> >
> > ---
> > Based on these additional results and clarifications, we hope you could consider increasing your score in support of this work. If not, could you kindly let us know what additionally needs to be done in your assessment to make this work ready for publication?

---

### Official Review · Reviewer_YjXg · 2025-11-02

**Soundness:** 2
**Presentation:** 3
**Contribution:** 2
**Rating:** 2
**Confidence:** 4

**Summary:**

The paper introduces ExpVid, a new benchmark designed to systematically evaluate Multimodal Large Language Models (MLLMs) on scientific experiment videos. The dataset consists of peer-reviewed laboratory videos from JoVE (an online journal), annotated with a three-level hierarchy: 1. Fine-grained Perception (e.g., identifying tools, materials, quantities, and actions); 2. Procedural Understanding (e.g., step ordering, sequence prediction, completeness verification); and 3. Scientific Reasoning (e.g., connecting experimental procedures to conclusions). The authors use multiple MLLMs (both open- and closed-source) to evaluate the performance on ExpVid. The results reveal a substantial gap between closed-source models (GPT-5, Gemini-2.5) and open-source counterparts in all tasks.

**Strengths:**

1. The authors introduce a dataset for the evaluation of MLLMs for scientific experimental reasoning, a domain previously underserved by benchmarks like MVBench, Video-MMMU, and SCI-VID. The three-level hierarchy (Perception -> Procedure -> Reasoning) mimics the actual cognitive workflow of scientific experimentation and it is interesting.
2. The annotation pipeline combines automated extraction by LLMs with expert human validation by PhD students.
3. Empirical evaluation with a large number of LLMs is seen as valuable.

**Weaknesses:**

1. Although an interesting dataset, using only one source to construct the dataset (JoVE) could introduce biases. It is unclear how the models would perform on other types of data (collected from other sources). Are there other sources that can be used to enhance the dataset? (Other journals with such data?)

2. The reliance of LLMs for annotation could potentially introduce some biases. Did the authors/PhD student annotators carefully look for such biases and ensure the dataset is properly annotated? A more thorough discussion on this would strengthen the annotation process.

3. The focus on wet-lab biology, chemistry, medicine could limit the scope of this paper and may not be of interest to a large audience in the ICLR community.

**Questions:**

1. Are there other domains where the construction of datasets in the same fashion can be expanded to ensure this work would target a larger audience from the ICLR community?

2. Can you please provide details on any biases that LLMs may have included in the annotation process (if any)?

---

> ### Author Response · Authors · 2025-11-21
>
> Thank you for your detailed feedback, which we believe will improve our manuscript. Please see our responses to the helpful points raised in your review below:
>
> ---
> >***W1** Single-source data (JoVE) could introduce biases*
>
> **A1** We appreciate the concern regarding the use of a single source. We selected JoVE because it is a peer-reviewed video journal where the manuscript and the recorded experiment video are co-submitted and jointly peer-reviewed. Each video produces step-level narrated ASR that aligns with the protocol, which is precisely the supervision our hierarchical annotations rely on. In contrast, most major journals do not require authors to submit recorded experiment videos, and most experiment videos from other sources lack such fine-grained, protocol-aligned textual grounding. This makes reliable large-scale video-to-language supervision infeasible for our benchmark. In the long term, academic resources such as JoVE will become increasingly crucial, driven by the scientific community's continuous pursuit of experimental reproducibility and comprehensive record-keeping. Additionally, MOOCs and university recordings are potential supplementary sources, but they usually lack the professional curation and step-level alignment needed for precise annotation. Substantial additional cleaning and expert annotation would be required to reach the same quality standard, where we consider this as a direction for future expansion.
>
> ---
> >***W2&Q2** Potential LLM biases in annotation & details for human annotation*
>
> **A2** We acknowledge the concern regarding potential LLM biases in the annotation pipeline. In our design, the LLM does not **create new factual content**; instead, it **extracts** key spans from high-quality textual sources: step-by-step narrated ASR or the paired paper. This extraction setup minimizes model-imposed bias.
> Nonetheless, automatic extraction may still introduce issues such as misinterpreting ASR sentences, generating counterfactual distractors, or minor ASR–video misalignment. To ensure reliability, every automatically produced item is verified by PhD-level human annotators, who check schema correctness, semantic fidelity, and alignment with the visual evidence. Only items that pass human verification are retained.
> Discussions on the criteria for eliminating such biases are provided as follows:
> - **Video-grounded**: All questions must be solvable from the visual evidence in the video.
> - **No leakage or guessing shortcuts**: Stems must not reveal answers; distractors must be scientifically plausible.
> - **Concrete, step-level fidelity**: Only visually verifiable actions are retained; abstract or unobservable descriptions are removed or rewritten.
> - **Consistent formatting and clarity**: Wording avoids unverifiable details and ensures each question has a unique, unambiguous answer.
> - **Justified verification**: Annotators provide a brief rationale for every accepted or corrected item to ensure transparency.
>
> We have added the aforementioned discussions and integrated the detailed Expert Verification Workflow into **Appendix E, page 20, of the revised manuscript**.

---

> ### Author Response · Authors · 2025-11-21
>
> >***W3** Wet-lab focus may narrow the benchmark’s scope and audience.*
>
> **A3** Thank you for raising this point regarding the scope and audience of our paper. We understand the concern that a focus on wet-lab disciplines may appear niche to a general ICLR audience. However, we respectfully argue that this focus is in fact a significant strength, as it directly addresses core challenges relevant to the broader machine learning community, particularly in areas like multimodal perception, reasoning, embodied AI, and AI for science.
> 1. **Broad Representation of Real-World Complexity**: We view wet-lab experimentation as a representative and high-impact setting for studying how MLLMs perceive, track, and reason about complex real-world procedural workflows. This setting is central to the rapidly growing field of AI-for-Science applications, which is gaining significant traction across various AI communities. The 13 wet-lab subfields included in our study, spanning biology, chemistry, and medicine, are not narrow silos. Instead, they collectively represent a remarkably broad range of real experimental environments. These environments are inherently characterized by complex, dynamic, and visually rich procedural workflows. Crucially, they demand multimodal perception, intricate step reasoning, and robust cross-modal grounding to correctly interpret and execute tasks—precisely the capabilities we aim to evaluate and advance in MLLMs.
> 2. **Distinction from Dry-Lab Simulations**: In contrast to many dry-lab "experiments", which are often simulation-based, wet-lab settings provide uniquely rich and visually grounded procedural signals. While dry-lab simulations are valuable for certain research questions, they typically lack nuanced, real-world interaction, visual ambiguities, and physical constraints inherent in actual laboratory work. Data from dry-lab simulations is often much easier to collect and annotate due to its controlled nature, but this very ease means it often doesn't capture the "hard core real-world interaction parts" that are critical for developing truly robust and generalizable AI systems. Our work is specifically designed to tackle these challenging, real-world interaction scenarios.
> 3. **Relevance to the ICLR Community**: The ICLR community has a strong interest in foundational AI capabilities, embodied intelligence, and real-world applications. Our benchmark provides a crucial foundation for researchers interested in applying embodied AI to automate complex real-world tasks, including scientific discovery. The "dirty jobs" of scientific experimentation, which involve intricate and often repetitive human operations, are ripe for AI assistance. To achieve this, we first need MLLMs that can correctly perceive and understand human exquisite operations in these demanding environments. Our work is therefore necessary to lay down the foundational capabilities required for future AI-driven scientific automation and discovery, making it highly relevant to researchers pushing the boundaries of AI in real-world contexts.
>
> In summary, while the domain is scientific wet-lab, the underlying AI challenges: multimodal understanding, complex reasoning, and real-world grounding, are universal and central to the interests of the ICLR community. We believe our benchmark offers a unique and challenging dataset that will accelerate research in these critical areas.
>
> ---
> >***Q1** Can construction fashion be expanded to other domains?*
>
> **A4** Yes. Our automated video-to-text construction pipeline could be applied to any procedural video domain that provides step-level textual accompaniment. Beyond wet labs, this includes areas such as electronics assembly/repair, physics and engineering laboratory procedures, and other settings where transcripts, text materials offer the same type of aligned side text needed to generate hierarchical annotations.
>
> ---
> In light of these clarifications, would you consider increasing your score on our paper? Otherwise, could you let us know any additional changes you would like to see in order for this work to be accepted?

---

### Meta-Review · Area_Chair_rcrp · 2026-01-05

**Summary:**

This paper received four positive scores and one negative score (Reviewer YjXg). The main concerns lie in (1) Biased single-source data and LLM notation bias, (2) Limited scope of the proposed benchmark in ICLR community, (3) The rationale for choosing “fill-in-the-blank” format for Level-3 tasks, (4) With/without thinking evaluation, (5) Small scale of videos, (6) Fail to evaluate the reasoning process, (7) Input Format Ambiguity & Clarification on Input Format, (7) Expert Annotation Scale and Feasibility, (8) The experimental details. The authors provided the corresponding experiments and analysis, which are well resolved according to the understanding of the Meta reviewer, despite no replies from all reviewers. In particular, it is satisfied regarding the response to Reviewer YjXg, where the notation bias and correction, and the rationale of the benchmark, affirm the contributions of the proposed ExpVid.

Overall, the proposed benchmark is interesting and novel, and the evaluation of multiple MLLMs on ExpVid is sufficient and comprehensive. Contributions are solid, offering a valuable roadmap for future research. After reading the paper, reviews, rebuttal, and the author's message, the Meta reviewer recommends accepting this paper.

**Reviewer Concerns:**

Most concerns are well addressed.

**Reviewer Scores:**

I think Reviewer YjXg would increase the score if they participated fully in the discussion.

---

### Decision · Program_Chairs · 2026-01-26

Accept (Poster)